# Wide Distribution of *Teratosphaeria epicoccoides* and *T. destructans* Associated with Diseased *Eucalyptus* Leaves in Plantations in Southern China

**DOI:** 10.3390/microorganisms12010129

**Published:** 2024-01-09

**Authors:** Bingyin Chen, Wenxia Wu, Shuaifei Chen

**Affiliations:** 1Research Institute of Fast-Growing Trees (RIFT), Chinese Academy of Forestry (CAF), Zhanjiang 524022, China; bingyinc@126.com (B.C.); wuwenxia_hainan@126.com (W.W.); 2College of Forestry, Nanjing Forestry University (NJFU), Nanjing 210037, China

**Keywords:** forest disease, genetic diversity, leaf pathogen, phylogeny, *Teratosphaeria* leaf blight, *Teratosphaeria* leaf spot

## Abstract

Species of *Mycosphaerellaceae* and *Teratosphaeriaceae* represent over 40% of the fungi identified on eucalypt leaves worldwide. These include some important pathogens that mainly cause leaf blight and spot, and result in increasingly negative impacts on global commercial eucalypt industries. *Eucalyptus* plantations are commonly cultivated in southern China for solid wood and pulp products. However, the species diversity and geographic distribution of *Mycosphaerellaceae* and *Teratosphaeriaceae*, associated with diseased plantation *Eucalyptus* leaves in China, have not been clarified. In this study, we conducted the first systematic surveys and sample collections of *Mycosphaerellaceae*- and *Teratosphaeriaceae*-like fungi from diseased plantation *Eucalyptus* leaves in southern China. In total, 558 isolates were obtained from 59 sampled sites in five provinces. One isolate was isolated from each tree. According to the disease symptoms, conidia morphological characteristics, and DNA sequence comparisons of ITS, *tef1* and *tub2* gene regions. The 558 isolates were identified as *Teratosphaeria epicoccoides* (312 isolates; 55.9%) and *T. destructans* (246 isolates, 44.1%). Both species were widely distributed in the sampled regions in southern China. The genotypes of *T. epicoccoides* and *T. destructans* were determined based on ITS, *tef1*, and *tub2* sequences. The results showed that multiple genotypes of each species of *T. epicoccoides* and *T. destructans* exist in China. Additionally, isolates with multiple genotypes were obtained in all five sampled provinces. These results suggest that both *T. epicoccoides* and *T. destructans* are not clonal. This study proved that both *T. epicoccoides* and *T. destructans* are dominant species and widely distributed on diseased *Eucalyptus* leaves in southern China. The wide geographic distribution and potential high genetic diversity pose challenges for the disease management of *Teratosphaeria* leaf blight and leaf spot in China.

## 1. Introduction

In China, eucalypts plantations have expanded rapidly to meet the increasing demand for wood and pulp [1]. Occupying approximately 5.4 million hm^2^ in China in 2018, the plantations account for 6.8% of the national plantation area and provide more than one-third of the total national commercial timber production [2]. *Eucalyptus* plantations are mainly planted in the southern regions of China, especially in Guangxi, Guangdong, Yunnan, and Fujian Provinces. Mostly, selected genotypes of *Eucalyptus urophylla* × *E. grandis* hybrids are planted and grown [3].

In China, *Eucalyptus* plantations are primarily located in tropical and subtropical regions, where a diverse array of pathogenic fungi can easily proliferate [4]. Due to extensive contiguous planting and a monotonous genetic makeup, *Eucalyptus* plantations in China are highly susceptible to the rapid and widespread proliferation of diseases when affected by some pathogenic microorganisms [5,6]. Currently, important diseases affecting plantation *Eucalyptus* in China include bacterial wilt, caused by *Ralstonia pseudosolanacearum* [6,7,8]; stem canker/wilt, caused by species of *Botryosphaeriaceae*, *Cryphonectriaceae*, and *Teratosphaeria zuluensis* [4,9,10,11,12]; leaf blight/spot, caused by species of *Teratosphaeriaceae* and *Mycosphaerellaceae* [13]; *Calonectria* [14,15,16,17]; and *Quambalaria* [18,19]. With the exception of bacterial wilt, the other diseases are all caused by pathogenic fungi.

Globally, leaf diseases on *Eucalyptus* caused by the fungi of *Mycosphaerellaceae* and *Teratosphaeriaceae* are widely distributed [20,21,22,23]. In recent years, a number of novel species of *Mycosphaerellaceae* and *Teratosphaeriaceae* have been isolated from diseased *Eucalyptus* leaves and described [23,24,25], some of which are known to cause leaf diseases on *Eucalyptus*. According to the research findings of Crous et al. [23], species belonging to *Mycosphaerellaceae* and *Teratosphaeriaceae* demonstrate a high species diversity on a global scale. Species from these two families represent over 40% of the fungi identified on eucalypts leaves worldwide [23].

In China, several species of *Mycosphaerellaceae* and *Teratosphaeriaceae* have been implicated as a cause of *Eucalyptus* leaf spot/blight [26,27,28,29,30]. Of the foliar pathogens belonging to *Mycosphaerellaceae* and *Teratosphaeriaceae* isolated from *Eucalyptus* leaves in China, only *T. destructans* (causing *Eucalyptus* leaf blight) [26] and *Pseudocercospora chiangmaiensis* (responsible for *Eucalyptus* circular leaf spot) [28,31] were identified based on both DNA sequence comparisons and morphological characteristics. The identification of the remaining species causing leaf spot/blight relied solely on morphological features [32,33,34,35,36,37]. More than 600 *Eucalyptus* species are native to Australia, with only a few species endemic to Papua New Guinea, some parts of Indonesia and the Philippines [38,39]. Many fungi of *Mycosphaerellaceae* and *Teratosphaeriaceae* have been described from *Eucalyptus* foliage in Australia, and most were considered to be endemic [23,40,41]. Previous studies have shown that *Mycosphaerellaceae* and *Teratosphaeriaceae* exhibit high species diversity; species in these two families are the predominant fungi on diseased *Eucalyptus* leaves globally [23]. In China, research on the species diversity of *Mycosphaerellaceae* and *Teratosphaeriaceae* fungi on diseased *Eucalyptus* leaves is very limited, and the identification of some of these species is not accurate because of the lack of DNA sequence data [26,27,29,30]. Furthermore, there are limited systematic studies on their geographical distribution. Recently, we collected diseased *Eucalyptus* leaves with typical mature fruiting structures of *Mycosphaerellaceae* and *Teratosphaeriaceae* from *Eucalyptus* plantations in Yunnan, Guangxi, Hainan, Guangdong, and Fujian Provinces in southern China. We subsequently isolated these fungi. The aims of this study were to: (i) identify these fungi based on DNA sequence comparisons of multi-gene regions and morphological characteristics; and (ii) explore the geographical distribution characteristics of *Mycosphaerellaceae* and *Teratosphaeriaceae* species on diseased *Eucalyptus* leaves in southern China.

## 2. Materials and Methods

### 2.1. Disease Symptoms, Samples, and Fungal Isolations

The study was mainly conducted from July to October 2022, and from February to April 2023. The sampled regions had high temperatures (20 °C–35 °C) and high levels of humidity (60–90%). Diseases caused by *Mycosphaerellaceae* and *Teratosphaeriaceae* were surveyed on *Eucalyptus* plantations in the Guangxi, Guangdong, Yunnan, Hainan, and Fujian provinces in southern China. At most sampling sites, the trees were 1 to 2 years old. *E. urophylla* × *E. grandis* hybrids were dominant, although a few genotypes of *E. urophylla*, *E. urophylla* × *E. tereticomis*, and *E. urophylla* × *E. pellita* were also surveyed (Table 1). In the plantations, the leaves of trees infected by *Mycosphaerellaceae* and *Teratosphaeriaceae* fungi, resulting in leaf spot, vein delimitation, chlorosis, and intense defoliation were identified (Figure 1A,B). In some surveyed regions, all trees in the plantations were infected (Figure 1A,B). Based on the diseased symptoms, two groups of fungi causing different diseases were observed. The first group of fungi mainly infected mature and old leaves and produced abundant small spots on the infected leaves. The second group of fungi mainly infected the juvenile leaves, but affected some mature leaves as well, and resulted in water-soaked, chlorosis, and wrinkled symptoms on the infected leaves (Figure 1C–N). Disease symptoms of the first group of fungi were frequently observed in most surveyed *Eucalyptus* plantations. Disease symptoms with the second group of fungi were observed occasionally in some regions in southern China (Figure 1O–R). Diseased leaves with typical fruiting structures of *Mycosphaerellaceae* and *Teratosphaeriaceae* were collected from 3–59 trees, or approximately 30 trees, at the majority of sampling sites, depending on the area of the sampled plantation. Diseased leaf samples were transported to the laboratory for morphological examination, isolation, and further assessments.

Fungal isolates in the conidiomata with the typical morphological characteristics of *Mycosphaerellaceae* and *Teratosphaeriaceae* [23] were isolated from diseased leaves. The conidia masses in the conidiomata were scattered onto 2% malt extract agar (MEA) (20 g malt extract powder and 20 g agar powder per liter of water; malt extract powder was obtained from the Beijing Shuangxuan Microbial Culture Medium Products factory, Beijing, China; the agar powder was obtained from Beijing Solarbio Science & Technology Co., Ltd., Beijing, China) with sterile needles under a stereoscopic microscope (AxioCam Stemi 2000C, Carl Zeiss, Jena, Germany). After incubation at 25 °C for 6–10 h, the germinated conidia were transferred individually onto fresh 2% MEA under the dissection microscope and incubated at 25 °C for four weeks to obtain single-conidium cultures. One single-conidium culture was obtained from leaves of each sampled tree. All the single-conidium cultures were deposited in the culture collection (CSF) of the Research Institute of Fast-growing Trees (RIFT), Chinese Academy of Forestry (CAF), in Zhanjiang, Guangdong Province, China.

### 2.2. DNA Extraction, PCR Amplification, and Sequencing

All isolates obtained were used for DNA extraction and sequence analyses. DNA was extracted from four-week-old cultures. Mycelia were scraped using a sterilized scalpel and transferred to 2.0 mL Eppendorf tubes. The total genomic DNA was extracted using the CTAB protocol, as described by van Burik et al. [42]. The extracted DNA was dissolved in 30 μL of TE buffer (1 M Tris-HCl and 0.5 M EDTA, pH = 8.0). To degrade the RNA, 2.5 μL RNase (10 mg/mL) was added at 37 °C for one hour. The DNA concentration was measured using a NanoDrop 2000 spectrometer (Thermo Fisher Scientific, Waltham, MA, USA).

Based on previous research results, DNA sequences of the internal transcribed spacer regions and intervening 5.8S nrRNA gene (ITS), partial translation elongation factor 1-alpha (*tef1*) gene, and partial β-tubulin (*tub2*) gene were used as reliable DNA barcodes to clearly distinguish species in *Mycosphaerellaceae* and *Teratosphaeriaceae*, especially species of *Teratosphaeria* [25,26,43,44]. The three partial gene regions of *T. epicoccoides* were amplified using the primer pairs V9G /ITS-4 [45], EF1-728F/EF2 [46,47] and BT2a/BT2b [48], respectively. The three partial gene regions of *T. destructans* were amplified using the primer pairs ITS-3 /ITS-4 [45], EF1-728F/EF1-986R [46] and BT2a/BT2b [48], respectively. The PCR procedure was conducted as described by Andjic and co-authors [26].

All the PCR products of all isolates obtained in this study were sequenced in both the forward and reverse directions with the same primers used in the PCR amplification. Sequence reactions were conducted at the Beijing Genomics Institute, Guangzhou, China. All sequences obtained were edited using MEGA v. 7.0 software [49], and were deposited in GenBank (https://www.ncbi.nlm.nih.gov, accessed on 6 December 2023). The ITS, *tef1*, and *tub2* gene regions were sequenced for all isolates obtained in this study.

### 2.3. Multi-Gene Phylogenetic Analyses and Species Identification

All isolates obtained in this study were genotyped by their ITS, *tef1*, and *tub2* sequences. Based on the genotypes generated by the ITS, *tef1*, and *tub2* sequences, sequences of two isolates for each ITS–*tef1*–*tub2* genotype were selected for phylogenetic analyses. Isolates with the same genotype were considered to be the same species.

The preliminary identities of the isolates obtained in this study were determined by conducting a standard nucleotide BLAST search using all the generated sequences of ITS, *tef1*, and *tub2*. The BLAST results indicated that the isolates obtained in this study were grouped in the genus *Teratosphaeria*. The sequences of the type strains closely related to the *Teratosphaeria* isolates sequenced in the current study were downloaded from the NCBI database (http://www.ncbi.nlm.nih.gov/ accessed on 15 October 2023) and used for phylogenetic analyses (Table 3).

**Table 2 microorganisms-12-00129-t002:** Isolates sequenced and used for phylogenetic analyses in this study.

Sampling Site No. ^a^	Province	Isolate No. ^b,c^	Sample and Isolate Information ^d^	Species	GenBank Accession No. ^e^			Genotype ^g^
					ITS	*tef1*	*tub2*	
1	Yunnan	CSF25654	20230315-2-(2)	*T. destructans*	OR961532	OR980975	OR973016	AAA
1	Yunnan	CSF25655	20230315-2-(3)	*T. destructans*	OR961533	OR980976	OR973017	AAA
1	Yunnan	CSF25656 ^c^	20230315-2-(4)	*T. destructans*	OR961534	OR980977	OR973018	BAA
1	Yunnan	CSF25657	20230315-2-(6)	*T. destructans*	OR961535	OR980978	OR973019	AAA
1	Yunnan	CSF25658	20230315-2-(7)	*T. destructans*	OR961536	OR980979	OR973020	AAA
1	Yunnan	CSF25659 ^c^	20230315-2-(8)	*T. epicoccoides*	OR961537	OR980980	OR973021	ABA
1	Yunnan	CSF25660	20230315-2-(9)	*T. destructans*	OR961538	OR980981	OR973022	AAA
1	Yunnan	CSF25661	20230315-2-(10)	*T. destructans*	OR961539	OR980982	OR973023	AAA
1	Yunnan	CSF25686	20230315-2-(12)	*T. epicoccoides*	OR961540	OR980983	OR973024	AAA
1	Yunnan	CSF25687	20230315-2-(13)	*T. epicoccoides*	OR961541	OR980984	OR973025	ECA
1	Yunnan	CSF25688	20230315-2-(15)	*T. epicoccoides*	OR961542	OR980985	OR973026	ECA
1	Yunnan	CSF25689	20230315-2-(16)	*T. epicoccoides*	OR961543	OR980986	OR973027	ACA
1	Yunnan	CSF25690	20230315-2-(17)	*T. epicoccoides*	OR961544	OR980987	OR973028	AAA
1	Yunnan	CSF25691	20230315-2-(19)	*T. epicoccoides*	OR961545	OR980988	OR973029	AAA
2	Yunnan	CSF25640	20230315-1-(1)	*T. destructans*	OR961546	OR980989	OR973030	AAA
2	Yunnan	CSF25641	20230315-1-(2)	*T. destructans*	OR961547	OR980990	OR973031	AAA
2	Yunnan	CSF25642	20230315-1-(3)	*T. destructans*	OR961548	OR980991	OR973032	AAA
2	Yunnan	CSF25643	20230315-1-(4)	*T. destructans*	OR961549	OR980992	OR973033	AAA
2	Yunnan	CSF25644	20230315-1-(5)	*T. destructans*	OR961550	OR980993	OR973034	AAA
2	Yunnan	CSF25645	20230315-1-(6)	*T. destructans*	OR961551	OR980994	OR973035	AAA
2	Yunnan	CSF25646	20230315-1-(7)	*T. destructans*	OR961552	OR980995	OR973036	AAA
2	Yunnan	CSF25647	20230315-1-(8)	*T. destructans*	OR961553	OR980996	OR973037	AAA
2	Yunnan	CSF25648	20230315-1-(9)	*T. destructans*	OR961554	OR980997	OR973038	AAA
2	Yunnan	CSF25649	20230315-1-(10)	*T. destructans*	OR961555	OR980998	OR973039	AAA
2	Yunnan	CSF25650	20230315-1-(11)	*T. destructans*	OR961556	OR980999	OR973040	AAA
2	Yunnan	CSF25651	20230315-1-(12)	*T. destructans*	OR961557	OR981000	OR973041	AAA
2	Yunnan	CSF25652	20230315-1-(13)	*T. destructans*	OR961558	OR981001	OR973042	AAA
2	Yunnan	CSF25653	20230315-1-(14)	*T. destructans*	OR961559	OR981002	OR973043	AAA
2	Yunnan	CSF25683 ^c^	20230315-1-(24)	*T. epicoccoides*	OR961560	OR981003	OR973044	AAA
2	Yunnan	CSF25685	20230315-1-(29)	*T. epicoccoides*	OR961561	OR981004	OR973045	AAA
3	Yunnan	CSF25671	20230314-2-(12)	*T. epicoccoides*	OR961562	OR981005	OR973046	AAA
3	Yunnan	CSF25672	20230314-2-(13)	*T. epicoccoides*	OR961563	OR981006	OR973047	ACA
3	Yunnan	CSF25673	20230314-2-(14)	*T. epicoccoides*	OR961564	OR981007	OR973048	AAA
3	Yunnan	CSF25674	20230314-2-(15)	*T. epicoccoides*	OR961565	OR981008	OR973049	EAA
3	Yunnan	CSF25675	20230314-2-(16)	*T. epicoccoides*	OR961566	OR981009	OR973050	AAA
3	Yunnan	CSF25676	20230314-2-(17)	*T. epicoccoides*	OR961567	OR981010	OR973051	AAA
4	Yunnan	CSF25637	20230314-1-(4)	*T. destructans*	OR961568	OR981011	OR973052	AAA
4	Yunnan	CSF25638	20230314-1-(10)	*T. destructans*	OR961569	OR981012	OR973053	AAA
4	Yunnan	CSF25639	20230314-1-(13)	*T. destructans*	OR961570	OR981013	OR973054	AAA
5	Yunnan	CSF25662	20230315-3-(1)	*T. destructans*	OR961571	OR981014	OR973055	AAA
5	Yunnan	CSF25663	20230315-3-(2)	*T. destructans*	OR961572	OR981015	OR973056	AAA
5	Yunnan	CSF25664	20230315-3-(3)	*T. destructans*	OR961573	OR981016	OR973057	AAA
5	Yunnan	CSF25665	20230315-3-(4)	*T. destructans*	OR961574	OR981017	OR973058	AAA
5	Yunnan	CSF25666	20230315-3-(5)	*T. destructans*	OR961575	OR981018	OR973059	AAA
5	Yunnan	CSF25667	20230315-3-(6)	*T. destructans*	OR961576	OR981019	OR973060	AAA
5	Yunnan	CSF25668	20230315-3-(7)	*T. destructans*	OR961577	OR981020	OR973061	AAA
5	Yunnan	CSF25669	20230315-3-(8)	*T. destructans*	OR961578	OR981021	OR973062	AAA
5	Yunnan	CSF25670	20230315-3-(9)	*T. destructans*	OR961579	OR981022	OR973063	AAA
5	Yunnan	CSF25698	20230315-3-(16)	*T. epicoccoides*	OR961580	OR981023	OR973064	AAA
5	Yunnan	CSF25699	20230315-3-(17)	*T. epicoccoides*	OR961581	OR981024	OR973065	ACA
5	Yunnan	CSF25700	20230315-3-(18)	*T. epicoccoides*	OR961582	OR981025	OR973066	EAA
5	Yunnan	CSF25701	20230315-3-(19)	*T. epicoccoides*	OR961583	OR981026	OR973067	EAA
5	Yunnan	CSF25702	20230315-3-(21)	*T. epicoccoides*	OR961584	OR981027	OR973068	ACA
6	Yunnan	CSF25703	20230316-2-(1)	*T. epicoccoides*	OR961585	OR981028	OR973069	DIA
6	Yunnan	CSF25704 ^c^	20230316-2-(2)	*T. epicoccoides*	OR961586	OR981029	OR973070	DIA
6	Yunnan	CSF25705	20230316-2-(4)	*T. epicoccoides*	OR961587	OR981030	OR973071	DIA
6	Yunnan	CSF25706	20230316-2-(6)	*T. epicoccoides*	OR961588	OR981031	OR973072	DIA
6	Yunnan	CSF25707	20230316-2-(7)	*T. epicoccoides*	OR961589	OR981032	OR973073	DIA
6	Yunnan	CSF25708	20230316-2-(8)	*T. epicoccoides*	OR961590	OR981033	OR973074	DIA
7	Yunnan	CSF25713	20230317-1-(1)	*T. epicoccoides*	OR961591	OR981034	OR973075	AAA
7	Yunnan	CSF25714	20230317-1-(2)	*T. epicoccoides*	OR961592	OR981035	OR973076	AAA
7	Yunnan	CSF25715	20230317-1-(4)	*T. epicoccoides*	OR961593	OR981036	OR973077	EAA
7	Yunnan	CSF25716	20230317-1-(6)	*T. epicoccoides*	OR961594	OR981037	OR973078	EAA
7	Yunnan	CSF25717	20230317-1-(7)	*T. epicoccoides*	OR961595	OR981038	OR973079	ECA
7	Yunnan	CSF25718	20230317-1-(8)	*T. epicoccoides*	OR961596	OR981039	OR973080	AAA
8	Yunnan	CSF25725	20230318-1-(1)	*T. epicoccoides*	OR961597	OR981040	OR973081	ECA
8	Yunnan	CSF25726	20230318-1-(2)	*T. epicoccoides*	OR961598	OR981041	OR973082	EAA
8	Yunnan	CSF25727 ^c^	20230318-1-(3)	*T. epicoccoides*	OR961599	OR981042	OR973083	FCA
8	Yunnan	CSF25728 ^c^	20230318-1-(4)	*T. epicoccoides*	OR961600	OR981043	OR973084	CAA
8	Yunnan	CSF25729	20230318-1-(5)	*T. epicoccoides*	OR961601	OR981044	OR973085	EAA
8	Yunnan	CSF25730	20230318-1-(6)	*T. epicoccoides*	OR961602	OR981045	OR973086	EAA
9	Yunnan	CSF25737	20230318-2-(1)	*T. epicoccoides*	OR961603	OR981046	OR973087	AAA
9	Yunnan	CSF25738	20230318-2-(2)	*T. epicoccoides*	OR961604	OR981047	OR973088	EAA
9	Yunnan	CSF25739	20230318-2-(3)	*T. epicoccoides*	OR961605	OR981048	OR973089	EAA
9	Yunnan	CSF25740	20230318-2-(4)	*T. epicoccoides*	OR961606	OR981049	OR973090	AAA
9	Yunnan	CSF25741	20230318-2-(5)	*T. epicoccoides*	OR961607	OR981050	OR973091	EAA
9	Yunnan	CSF25742	20230318-2-(6)	*T. epicoccoides*	OR961608	OR981051	OR973092	EAA
10	Yunnan	CSF25750	20230318-3-(1)	*T. epicoccoides*	OR961609	OR981052	OR973093	EAA
10	Yunnan	CSF25751	20230318-3-(2)	*T. epicoccoides*	OR961610	OR981053	OR973094	EAA
10	Yunnan	CSF25752	20230318-3-(3)	*T. epicoccoides*	OR961611	OR981054	OR973095	EAA
10	Yunnan	CSF25753	20230318-3-(4)	*T. epicoccoides*	OR961612	OR981055	OR973096	CAA
10	Yunnan	CSF25754	20230318-3-(7)	*T. epicoccoides*	OR961613	OR981056	OR973097	AAA
10	Yunnan	CSF25755	20230318-3-(8)	*T. epicoccoides*	OR961614	OR981057	OR973098	AAA
11	Yunnan	CSF25763	20230319-1-(2)	*T. epicoccoides*	OR961615	OR981058	OR973099	EAA
11	Yunnan	CSF25764	20230319-1-(3)	*T. epicoccoides*	OR961616	OR981059	OR973100	ACA
11	Yunnan	CSF25765	20230319-1-(5)	*T. epicoccoides*	OR961617	OR981060	OR973101	AAA
11	Yunnan	CSF25766	20230319-1-(6)	*T. epicoccoides*	OR961618	OR981061	OR973102	CAA
11	Yunnan	CSF25767	20230319-1-(7)	*T. epicoccoides*	OR961619	OR981062	OR973103	AAA
12	Guangxi	CSF25066 ^c^	20220703-1-(1)	*T. epicoccoides*	OR961620	OR981063	OR973104	ABA
12	Guangxi	CSF25067	20220703-1-(3)	*T. destructans*	OR961621	OR981064	OR973105	CAA
12	Guangxi	CSF25068	20220703-1-(4)	*T. destructans*	OR961622	OR981065	OR973106	CAA
12	Guangxi	CSF25069	20220703-1-(5)	*T. destructans*	OR961623	OR981066	OR973107	AAA
12	Guangxi	CSF25070	20220703-1-(6)	*T. destructans*	OR961624	OR981067	OR973108	CAA
12	Guangxi	CSF25071	20220703-1-(8)	*T. destructans*	OR961625	OR981068	OR973109	AAA
12	Guangxi	CSF25072 ^c^	20220703-1-(10)	*T. epicoccoides*	OR961626	OR981069	OR973110	XDA
12	Guangxi	CSF25218 ^c^	20220703-1-(12)	*T. epicoccoides*	OR961627	OR981070	OR973111	AAA
12	Guangxi	CSF25219	20220703-1-(14)	*T. epicoccoides*	OR961628	OR981071	OR973112	CAA
12	Guangxi	CSF25220	20220703-1-(15)	*T. epicoccoides*	OR961629	OR981072	OR973113	ECA
12	Guangxi	CSF25222	20220703-1-(18)	*T. epicoccoides*	OR961630	OR981073	OR973114	CAA
12	Guangxi	CSF25223	20220703-1-(19)	*T. epicoccoides*	OR961631	OR981074	OR973115	BAA
13	Guangxi	CSF25083	20220703-3-(1)	*T. destructans*	OR961632	OR981075	OR973116	AAA
13	Guangxi	CSF25084	20220703-3-(2)	*T. destructans*	OR961633	OR981076	OR973117	CAA
13	Guangxi	CSF25085	20220703-3-(3)	*T. destructans*	OR961634	OR981077	OR973118	AAA
13	Guangxi	CSF25086	20220703-3-(4)	*T. destructans*	OR961635	OR981078	OR973119	CAA
13	Guangxi	CSF25087	20220703-3-(5)	*T. destructans*	OR961636	OR981079	OR973120	CAA
13	Guangxi	CSF25088	20220703-3-(6)	*T. epicoccoides*	OR961637	OR981080	OR973121	XBA
13	Guangxi	CSF25089	20220703-3-(7)	*T. destructans*	OR961638	OR981081	OR973122	AAA
13	Guangxi	CSF25090	20220703-3-(9)	*T. destructans*	OR961639	OR981082	OR973123	CAA
13	Guangxi	CSF25091	20220703-3-(10)	*T. destructans*	OR961640	OR981083	OR973124	AAA
13	Guangxi	CSF25240 ^c^	20220703-3-(17)	*T. epicoccoides*	OR961641	OR981084	OR973125	EAA
13	Guangxi	CSF25241	20220703-3-(19)	*T. epicoccoides*	OR961642	OR981085	OR973126	EAA
13	Guangxi	CSF25242	20220703-3-(20)	*T. epicoccoides*	OR961643	OR981086	OR973127	EAA
13	Guangxi	CSF25243	20220703-3-(22)	*T. epicoccoides*	OR961644	OR981087	OR973128	ACA
13	Guangxi	CSF25244	20220703-3-(23)	*T. epicoccoides*	OR961645	OR981088	OR973129	AAA
13	Guangxi	CSF25245	20220703-3-(24)	*T. epicoccoides*	OR961646	OR981089	OR973130	ACA
14	Guangxi	CSF25073 ^c^	20220703-2-(1)	*T. destructans*	OR961647	OR981090	OR973131	AAA
14	Guangxi	CSF25074	20220703-2-(2)	*T. destructans*	OR961648	OR981091	OR973132	AAA
14	Guangxi	CSF25075 ^c^	20220703-2-(3)	*T. epicoccoides*	OR961649	OR981092	OR973133	XBA
14	Guangxi	CSF25076	20220703-2-(4)	*T. destructans*	OR961650	OR981093	OR973134	AAA
14	Guangxi	CSF25077	20220703-2-(5)	*T. destructans*	OR961651	OR981094	OR973135	AAA
14	Guangxi	CSF25078	20220703-2-(6)	*T. destructans*	OR961652	OR981095	OR973136	AAA
14	Guangxi	CSF25079	20220703-2-(7)	*T. epicoccoides*	OR961653	OR981096	OR973137	ABA
14	Guangxi	CSF25080	20220703-2-(8)	*T. destructans*	OR961654	OR981097	OR973138	CAA
14	Guangxi	CSF25081	20220703-2-(9)	*T. epicoccoides*	OR961655	OR981098	OR973139	XBA
14	Guangxi	CSF25082	20220703-2-(10)	*T. destructans*	OR961656	OR981099	OR973140	CAA
14	Guangxi	CSF25231	20220703-2-(21)	*T. epicoccoides*	OR961657	OR981100	OR973141	AAA
14	Guangxi	CSF25232	20220703-2-(22)	*T. epicoccoides*	OR961658	OR981101	OR973142	AAA
14	Guangxi	CSF25233	20220703-2-(25)	*T. epicoccoides*	OR961659	OR981102	OR973143	EAA
15	Guangxi	CSF25092	20220703-4-(1)	*T. destructans*	OR961660	OR981103	OR973144	AAA
15	Guangxi	CSF25093	20220703-4-(3)	*T. destructans*	OR961661	OR981104	OR973145	AAA
15	Guangxi	CSF25094	20220703-4-(4)	*T. destructans*	OR961662	OR981105	OR973146	CAA
15	Guangxi	CSF25095	20220703-4-(5)	*T. destructans*	OR961663	OR981106	OR973147	CAA
15	Guangxi	CSF25096	20220703-4-(6)	*T. destructans*	OR961664	OR981107	OR973148	AAA
15	Guangxi	CSF25097	20220703-4-(7)	*T. destructans*	OR961665	OR981108	OR973149	AAA
15	Guangxi	CSF25098	20220703-4-(8)	*T. destructans*	OR961666	OR981109	OR973150	CAA
15	Guangxi	CSF25099	20220703-4-(9)	*T. destructans*	OR961667	OR981110	OR973151	CAA
15	Guangxi	CSF25247 ^c^	20220703-4-(10)	*T. epicoccoides*	OR961668	OR981111	OR973152	ECA
15	Guangxi	CSF25248	20220703-4-(11)	*T. epicoccoides*	OR961669	OR981112	OR973153	AAA
16	Guangxi	CSF25056	20220702-3-(1)	*T. destructans*	OR961670	OR981113	OR973154	CAA
16	Guangxi	CSF25057	20220702-3-(2)	*T. destructans*	OR961671	OR981114	OR973155	CAA
16	Guangxi	CSF25058	20220702-3-(3)	*T. destructans*	– ^f^	OR981115	OR973156	-AA
16	Guangxi	CSF25059	20220702-3-(4)	*T. destructans*	OR961672	OR981116	OR973157	CAA
16	Guangxi	CSF25060	20220702-3-(5)	*T. destructans*	OR961673	OR981117	OR973158	AAA
16	Guangxi	CSF25061	20220702-3-(6)	*T. destructans*	OR961674	OR981118	OR973159	CAA
16	Guangxi	CSF25062	20220702-3-(7)	*T. destructans*	OR961675	OR981119	OR973160	AAA
16	Guangxi	CSF25063	20220702-3-(8)	*T. destructans*	OR961676	OR981120	OR973161	CAA
16	Guangxi	CSF25064	20220702-3-(9)	*T. destructans*	OR961677	OR981121	OR973162	CAA
16	Guangxi	CSF25065	20220702-3-(10)	*T. destructans*	OR961678	OR981122	OR973163	CAA
16	Guangxi	CSF25214 ^c^	20220702-3-(17)	*T. destructans*	OR961679	OR981123	OR973164	CBA
16	Guangxi	CSF25215	20220702-3-(18)	*T. epicoccoides*	OR961680	OR981124	OR973165	ECA
16	Guangxi	CSF25216	20220702-3-(19)	*T. epicoccoides*	OR961681	OR981125	OR973166	ECA
16	Guangxi	CSF25217	20220702-3-(20)	*T. epicoccoides*	OR961682	OR981126	OR973167	EAA
17	Guangxi	CSF25046	20220702-2-(1)	*T. destructans*	OR961683	OR981127	OR973168	CAA
17	Guangxi	CSF25047	20220702-2-(2)	*T. destructans*	OR961684	OR981128	OR973169	AAA
17	Guangxi	CSF25048 ^c^	20220702-2-(3)	*T. epicoccoides*	OR961685	OR981129	OR973170	CBA
17	Guangxi	CSF25049	20220702-2-(4)	*T. destructans*	OR961686	OR981130	OR973171	AAA
17	Guangxi	CSF25050	20220702-2-(5)	*T. destructans*	OR961687	OR981131	OR973172	AAA
17	Guangxi	CSF25051	20220702-2-(6)	*T. destructans*	OR961688	OR981132	OR973173	CAA
17	Guangxi	CSF25052	20220702-2-(7)	*T. destructans*	OR961689	OR981133	OR973174	AAA
17	Guangxi	CSF25053	20220702-2-(8)	*T. destructans*	OR961690	OR981134	OR973175	AAA
17	Guangxi	CSF25054 ^c^	20220702-2-(9)	*T. destructans*	OR961691	OR981135	OR973176	CAA
17	Guangxi	CSF25055	20220702-2-(10)	*T. destructans*	OR961692	OR981136	OR973177	AAA
17	Guangxi	CSF25204	20220702-2-(17)	*T. epicoccoides*	OR961693	OR981137	OR973178	EAA
17	Guangxi	CSF25205	20220702-2-(18)	*T. epicoccoides*	OR961694	OR981138	OR973179	EAA
17	Guangxi	CSF25206	20220702-2-(19)	*T. epicoccoides*	OR961695	OR981139	OR973180	EAA
17	Guangxi	CSF25207	20220702-2-(20)	*T. epicoccoides*	OR961696	OR981140	OR973181	ECA
18	Guangxi	CSF25036	20220702-1-(1)	*T. destructans*	OR961697	OR981141	–	AA-
18	Guangxi	CSF25037	20220702-1-(2)	*T. destructans*	OR961698	OR981142	OR973182	AAA
18	Guangxi	CSF25038	20220702-1-(3)	*T. destructans*	OR961699	OR981143	OR973183	CAA
18	Guangxi	CSF25039	20220702-1-(4)	*T. destructans*	OR961700	OR981144	OR973184	AAA
18	Guangxi	CSF25040	20220702-1-(5)	*T. destructans*	OR961701	OR981145	OR973185	AAA
18	Guangxi	CSF25041	20220702-1-(6)	*T. destructans*	OR961702	OR981146	OR973186	AAA
18	Guangxi	CSF25042 ^c^	20220702-1-(7)	*T. epicoccoides*	OR961703	OR981147	OR973187	DJA
18	Guangxi	CSF25043 ^c^	20220702-1-(8)	*T. epicoccoides*	OR961704	OR981148	OR973188	AHA
18	Guangxi	CSF25044 ^c^	20220702-1-(9)	*T. epicoccoides*	OR961705	OR981149	OR973189	DJA
18	Guangxi	CSF25045 ^c^	20220702-1-(10)	*T. epicoccoides*	OR961706	OR981150	OR973190	DHA
18	Guangxi	CSF25188 ^c^	20220702-1-(11)	*T. epicoccoides*	OR961707	OR981151	OR973191	ACA
18	Guangxi	CSF25189	20220702-1-(12)	*T. epicoccoides*	OR961708	OR981152	OR973192	ACA
18	Guangxi	CSF25192	20220702-1-(15)	*T. epicoccoides*	OR961709	OR981153	OR973193	ACA
18	Guangxi	CSF25196	20220702-1-(19)	*T. epicoccoides*	OR961710	OR981154	OR973194	AEA
19	Guangxi	CSF25100	20220704-1-(1)	*T. destructans*	OR961711	OR981155	OR973195	CAA
19	Guangxi	CSF25101	20220704-1-(2)	*T. destructans*	OR961712	OR981156	OR973196	CAA
19	Guangxi	CSF25102	20220704-1-(3)	*T. destructans*	OR961713	OR981157	OR973197	CCA
19	Guangxi	CSF25103	20220704-1-(4)	*T. destructans*	OR961714	OR981158	OR973198	AAA
19	Guangxi	CSF25104	20220704-1-(5)	*T. destructans*	OR961715	OR981159	OR973199	CAA
19	Guangxi	CSF25105	20220704-1-(6)	*T. destructans*	OR961716	OR981160	OR973200	CAA
19	Guangxi	CSF25106	20220704-1-(7)	*T. destructans*	OR961717	OR981161	OR973201	CAA
19	Guangxi	CSF25107	20220704-1-(8)	*T. destructans*	OR961718	OR981162	OR973202	CAA
19	Guangxi	CSF25108	20220704-1-(12)	*T. destructans*	OR961719	OR981163	OR973203	CAA
19	Guangxi	CSF25109	20220704-1-(13)	*T. destructans*	OR961720	OR981164	OR973204	CAA
19	Guangxi	CSF25250	20220704-1-(17)	*T. epicoccoides*	OR961721	OR981165	OR973205	ECA
19	Guangxi	CSF25251 ^c^	20220704-1-(18)	*T. epicoccoides*	OR961722	OR981166	OR973206	CCA
19	Guangxi	CSF25252	20220704-1-(19)	*T. epicoccoides*	OR961723	OR981167	OR973207	ACA
19	Guangxi	CSF25253 ^c^	20220704-1-(20)	*T. epicoccoides*	OR961724	OR981168	OR973208	GGA
19	Guangxi	CSF25254	20220704-1-(21)	*T. epicoccoides*	OR961725	OR981169	OR973209	ACA
20	Guangxi	CSF25110	20220704-2-(1)	*T. destructans*	OR961726	OR981170	OR973210	CAA
20	Guangxi	CSF25111	20220704-2-(2)	*T. destructans*	OR961727	OR981171	OR973211	AAA
20	Guangxi	CSF25112	20220704-2-(4)	*T. destructans*	OR961728	OR981172	OR973212	AAA
20	Guangxi	CSF25113	20220704-2-(5)	*T. destructans*	OR961729	OR981173	OR973213	AAA
20	Guangxi	CSF25114	20220704-2-(6)	*T. destructans*	OR961730	OR981174	OR973214	CAA
20	Guangxi	CSF25115	20220704-2-(7)	*T. destructans*	OR961731	OR981175	OR973215	AAA
20	Guangxi	CSF25116	20220704-2-(8)	*T. destructans*	OR961732	OR981176	OR973216	CAA
20	Guangxi	CSF25117	20220704-2-(10)	*T. destructans*	OR961733	OR981177	OR973217	AAA
20	Guangxi	CSF25118	20220704-2-(11)	*T. destructans*	OR961734	OR981178	OR973218	AAA
20	Guangxi	CSF25119	20220704-2-(12)	*T. destructans*	OR961735	OR981179	OR973219	AAA
20	Guangxi	CSF25120	20220704-2-(13)	*T. destructans*	OR961736	OR981180	OR973220	CAA
20	Guangxi	CSF25121	20220704-2-(14)	*T. destructans*	OR961737	OR981181	OR973221	ACA
20	Guangxi	CSF25122	20220704-2-(15)	*T. destructans*	OR961738	OR981182	OR973222	AAA
20	Guangxi	CSF25123	20220704-2-(16)	*T. destructans*	OR961739	OR981183	OR973223	CAA
20	Guangxi	CSF25258	20220704-2-(17)	*T. epicoccoides*	OR961740	OR981184	OR973224	DIA
20	Guangxi	CSF25259	20220704-2-(18)	*T. epicoccoides*	OR961741	OR981185	OR973225	ACA
20	Guangxi	CSF25260	20220704-2-(19)	*T. epicoccoides*	OR961742	OR981186	OR973226	AFA
20	Guangxi	CSF25261	20220704-2-(20)	*T. epicoccoides*	OR961743	OR981187	OR973227	ACA
20	Guangxi	CSF25262	20220704-2-(21)	*T. epicoccoides*	OR961744	OR981188	OR973228	ACA
20	Guangxi	CSF25263	20220704-2-(22)	*T. epicoccoides*	OR961745	OR981189	OR973229	ACA
21	Guangxi	CSF25124	20220704-3-(1)	*T. destructans*	OR961746	OR981190	OR973230	CAA
21	Guangxi	CSF25125	20220704-3-(2)	*T. destructans*	OR961747	OR981191	OR973231	CAA
21	Guangxi	CSF25126	20220704-3-(3)	*T. destructans*	OR961748	OR981192	OR973232	CAA
21	Guangxi	CSF25127	20220704-3-(4)	*T. destructans*	OR961749	OR981193	OR973233	CAA
21	Guangxi	CSF25128	20220704-3-(5)	*T. destructans*	–	–	OR973234	--A
21	Guangxi	CSF25129	20220704-3-(6)	*T. destructans*	OR961750	OR981194	OR973235	CAA
21	Guangxi	CSF25130	20220704-3-(7)	*T. destructans*	OR961751	OR981195	OR973236	CAA
21	Guangxi	CSF25131	20220704-3-(8)	*T. destructans*	OR961752	OR981196	OR973237	CAA
21	Guangxi	CSF25132	20220704-3-(9)	*T. destructans*	OR961753	OR981197	OR973238	AAA
21	Guangxi	CSF25133	20220704-3-(10)	*T. destructans*	OR961754	OR981198	OR973239	CAA
21	Guangxi	CSF25270	20220704-3-(11)	*T. epicoccoides*	OR961755	OR981199	OR973240	ACA
21	Guangxi	CSF25271	20220704-3-(13)	*T. epicoccoides*	OR961756	OR981200	OR973241	EAA
21	Guangxi	CSF25272	20220704-3-(14)	*T. epicoccoides*	OR961757	OR981201	OR973242	EAA
21	Guangxi	CSF25273	20220704-3-(16)	*T. epicoccoides*	OR961758	OR981202	OR973243	ACA
21	Guangxi	CSF25274	20220704-3-(18)	*T. epicoccoides*	OR961759	–	OR973244	A-A
21	Guangxi	CSF25275	20220704-3-(20)	*T. epicoccoides*	OR961760	OR981203	OR973245	EAA
22	Hainan	CSF25309	20221026-6-(1)	*T. epicoccoides*	OR961761	OR981204	OR973246	ECA
22	Hainan	CSF25310	20221026-6-(2)	*T. epicoccoides*	OR961762	OR981205	OR973247	ECA
22	Hainan	CSF25311	20221026-6-(3)	*T. epicoccoides*	OR961763	OR981206	OR973248	EAA
23	Hainan	CSF25158	20221026-7-(1)	*T. destructans*	OR961764	OR981207	OR973249	CAA
23	Hainan	CSF25159	20221026-7-(2)	*T. destructans*	OR961765	OR981208	OR973250	CAA
23	Hainan	CSF25160	20221026-7-(3)	*T. destructans*	OR961766	OR981209	OR973251	CAA
23	Hainan	CSF25161	20221026-7-(4)	*T. destructans*	OR961767	OR981210	OR973252	AAA
23	Hainan	CSF25162	20221026-7-(5)	*T. destructans*	OR961768	OR981211	OR973253	CAA
23	Hainan	CSF25163	20221026-7-(6)	*T. destructans*	OR961769	OR981212	OR973254	AAA
23	Hainan	CSF25164	20221026-7-(7)	*T. destructans*	OR961770	OR981213	OR973255	AAA
23	Hainan	CSF25165	20221026-7-(8)	*T. destructans*	OR961771	OR981214	OR973256	AAA
23	Hainan	CSF25166	20221026-7-(9)	*T. destructans*	OR961772	OR981215	OR973257	AAA
23	Hainan	CSF25167	20221026-7-(10)	*T. destructans*	OR961773	OR981216	OR973258	CAA
23	Hainan	CSF25168	20221026-7-(11)	*T. destructans*	OR961774	OR981217	OR973259	CAA
23	Hainan	CSF25312	20221026-7-(12)	*T. epicoccoides*	OR961775	OR981218	OR973260	ACA
23	Hainan	CSF25313	20221026-7-(13)	*T. epicoccoides*	OR961776	OR981219	OR973261	ACA
23	Hainan	CSF25314	20221026-7-(14)	*T. epicoccoides*	OR961777	OR981220	OR973262	ACA
23	Hainan	CSF25315	20221026-7-(15)	*T. epicoccoides*	OR961778	OR981221	OR973263	DIA
23	Hainan	CSF25316	20221026-7-(16)	*T. epicoccoides*	OR961779	OR981222	OR973264	ACA
23	Hainan	CSF25317 ^c^	20221026-7-(17)	*T. epicoccoides*	OR961780	OR981223	OR973265	AIA
24	Hainan	CSF25144	20221026-4-(1)	*T. destructans*	OR961781	OR981224	OR973266	CAA
24	Hainan	CSF25145	20221026-4-(2)	*T. destructans*	OR961782	OR981225	OR973267	CAA
24	Hainan	CSF25146	20221026-4-(3)	*T. destructans*	OR961783	OR981226	OR973268	AAA
24	Hainan	CSF25147	20221026-4-(4)	*T. destructans*	OR961784	OR981227	OR973269	AAA
24	Hainan	CSF25292 ^c^	20221026-4-(5)	*T. epicoccoides*	OR961785	OR981228	OR973270	GGA
24	Hainan	CSF25293 ^c^	20221026-4-(6)	*T. epicoccoides*	OR961786	OR981229	OR973271	AFA
24	Hainan	CSF25294	20221026-4-(7)	*T. epicoccoides*	–	OR981230	OR973272	-EA
24	Hainan	CSF25295 ^c^	20221026-4-(8)	*T. epicoccoides*	OR961787	OR981231	OR973273	AEA
24	Hainan	CSF25296	20221026-4-(9)	*T. epicoccoides*	OR961788	OR981232	OR973274	EAA
24	Hainan	CSF25297	20221026-4-(10)	*T. epicoccoides*	OR961789	OR981233	OR973275	DIA
25	Hainan	CSF25148	20221026-5-(1)	*T. destructans*	OR961790	OR981234	OR973276	CAA
25	Hainan	CSF25149	20221026-5-(2)	*T. destructans*	OR961791	OR981235	OR973277	AAA
25	Hainan	CSF25150	20221026-5-(3)	*T. destructans*	OR961792	OR981236	OR973278	AAA
25	Hainan	CSF25151	20221026-5-(4)	*T. destructans*	OR961793	OR981237	OR973279	CAA
25	Hainan	CSF25152	20221026-5-(5)	*T. destructans*	OR961794	OR981238	OR973280	CAA
25	Hainan	CSF25153	20221026-5-(6)	*T. destructans*	OR961795	OR981239	OR973281	AAA
25	Hainan	CSF25154	20221026-5-(7)	*T. destructans*	OR961796	OR981240	OR973282	ACA
25	Hainan	CSF25155	20221026-5-(8)	*T. destructans*	OR961797	OR981241	OR973283	CAA
25	Hainan	CSF25156	20221026-5-(9)	*T. destructans*	OR961798	–	OR973284	C-A
25	Hainan	CSF25157	20221026-5-(10)	*T. destructans*	OR961799	OR981242	OR973285	AAA
25	Hainan	CSF25299 ^c^	20221026-5-(11)	*T. epicoccoides*	OR961800	OR981243	OR973286	AEA
25	Hainan	CSF25300	20221026-5-(12)	*T. epicoccoides*	OR961801	OR981244	OR973287	AEA
25	Hainan	CSF25301	20221026-5-(13)	*T. epicoccoides*	OR961802	OR981245	OR973288	AIA
25	Hainan	CSF25302 ^c^	20221026-5-(14)	*T. epicoccoides*	OR961803	OR981246	OR973289	AFA
25	Hainan	CSF25303	20221026-5-(15)	*T. epicoccoides*	OR961804	OR981247	OR973290	AAA
26	Hainan	CSF25134 ^c^	20221026-2-(1)	*T. destructans*	OR961805	OR981248	OR973291	CCA
26	Hainan	CSF25135	20221026-2-(2)	*T. destructans*	OR961806	OR981249	OR973292	ACA
26	Hainan	CSF25136	20221026-2-(3)	*T. destructans*	OR961807	OR981250	OR973293	ACA
26	Hainan	CSF25286	20221026-2-(4)	*T. epicoccoides*	–	OR981251	OR973294	-EA
26	Hainan	CSF25287	20221026-2-(5)	*T. epicoccoides*	OR961808	OR981252	OR973295	CAA
26	Hainan	CSF25288	20221026-2-(6)	*T. epicoccoides*	OR961809	OR981253	OR973296	AAA
26	Hainan	CSF25289	20221026-2-(7)	*T. epicoccoides*	–	OR981254	OR973297	-IA
26	Hainan	CSF25290	20221026-2-(8)	*T. epicoccoides*	OR961810	OR981255	–	AE-
26	Hainan	CSF25291	20221026-2-(9)	*T. epicoccoides*	–	OR981256	OR973298	-EA
27	Hainan	CSF25137 ^c^	20221026-3-(1)	*T. destructans*	OR961811	OR981257	OR973299	ACA
27	Hainan	CSF25138	20221026-3-(2)	*T. destructans*	OR961812	OR981258	OR973300	CAA
27	Hainan	CSF25139	20221026-3-(3)	*T. destructans*	OR961813	OR981259	OR973301	AAA
27	Hainan	CSF25140	20221026-3-(4)	*T. destructans*	OR961814	OR981260	OR973302	AAA
27	Hainan	CSF25141	20221026-3-(5)	*T. destructans*	OR961815	OR981261	OR973303	AAA
27	Hainan	CSF25142	20221026-3-(6)	*T. destructans*	OR961816	OR981262	OR973304	CAA
27	Hainan	CSF25143	20221026-3-(7)	*T. destructans*	OR961817	OR981263	OR973305	CAA
28	Hainan	CSF25276	20221026-1-(51)	*T. epicoccoides*	OR961818	OR981264	OR973306	EAA
28	Hainan	CSF25277 ^c^	20221026-1-(52)	*T. epicoccoides*	OR961819	OR981265	OR973307	DEA
28	Hainan	CSF25280	20221026-1-(55)	*T. epicoccoides*	OR961820	OR981266	OR973308	BAA
28	Hainan	CSF25283	20221026-1-(58)	*T. epicoccoides*	OR961821	OR981267	OR973309	AAA
28	Hainan	CSF25284	20221026-1-(59)	*T. epicoccoides*	OR961822	OR981268	OR973310	CAA
29	Guangdong	CSF25612	20230221-2-(1)	*T. destructans*	OR961823	OR981269	OR973311	CAA
29	Guangdong	CSF25613	20230221-2-(2)	*T. destructans*	OR961824	OR981270	OR973312	CAA
29	Guangdong	CSF25614	20230221-2-(3)	*T. destructans*	OR961825	OR981271	OR973313	ACA
29	Guangdong	CSF25615	20230221-2-(4)	*T. destructans*	OR961826	OR981272	OR973314	CAA
29	Guangdong	CSF25616	20230221-2-(5)	*T. destructans*	OR961827	OR981273	OR973315	AAA
29	Guangdong	CSF25617	20230221-2-(6)	*T. destructans*	OR961828	OR981274	OR973316	AAA
29	Guangdong	CSF25618	20230221-2-(7)	*T. destructans*	OR961829	OR981275	OR973317	CAA
29	Guangdong	CSF25619	20230221-2-(8)	*T. destructans*	OR961830	OR981276	OR973318	CAA
29	Guangdong	CSF25620	20230221-2-(9)	*T. destructans*	OR961831	OR981277	OR973319	CAA
29	Guangdong	CSF25621	20230221-2-(10)	*T. destructans*	OR961832	OR981278	OR973320	ACA
29	Guangdong	CSF25622	20230221-2-(11)	*T. destructans*	OR961833	OR981279	OR973321	CAA
29	Guangdong	CSF25623	20230221-2-(12)	*T. destructans*	OR961834	OR981280	OR973322	AAA
29	Guangdong	CSF25520	20230221-2-(20)	*T. epicoccoides*	OR961835	OR981281	OR973323	EAA
29	Guangdong	CSF25521	20230221-2-(21)	*T. epicoccoides*	OR961836	OR981282	OR973324	ACA
29	Guangdong	CSF25522	20230221-2-(22)	*T. epicoccoides*	OR961837	OR981283	OR973325	AAA
29	Guangdong	CSF25523	20230221-2-(23)	*T. epicoccoides*	OR961838	OR981284	OR973326	EAA
29	Guangdong	CSF25524	20230221-2-(25)	*T. epicoccoides*	OR961839	OR981285	OR973327	EAA
29	Guangdong	CSF25525	20230221-2-(26)	*T. epicoccoides*	OR961840	OR981286	OR973328	EAA
30	Guangdong	CSF25599	20230221-1-(1)	*T. destructans*	OR961841	OR981287	OR973329	CAA
30	Guangdong	CSF25600	20230221-1-(2)	*T. destructans*	OR961842	OR981288	OR973330	AAA
30	Guangdong	CSF25601 ^c^	20230221-1-(3)	*T. destructans*	OR961843	OR981289	OR973331	CCA
30	Guangdong	CSF25602	20230221-1-(4)	*T. destructans*	OR961844	OR981290	OR973332	AAA
30	Guangdong	CSF25603	20230221-1-(5)	*T. destructans*	OR961845	OR981291	OR973333	AAA
30	Guangdong	CSF25604	20230221-1-(6)	*T. destructans*	OR961846	OR981292	OR973334	AAA
30	Guangdong	CSF25605	20230221-1-(7)	*T. destructans*	OR961847	OR981293	OR973335	CAA
30	Guangdong	CSF25606	20230221-1-(8)	*T. destructans*	OR961848	OR981294	OR973336	AAA
30	Guangdong	CSF25607	20230221-1-(9)	*T. destructans*	OR961849	OR981295	OR973337	ACA
30	Guangdong	CSF25608	20230221-1-(10)	*T. destructans*	OR961850	OR981296	OR973338	AAA
30	Guangdong	CSF25609	20230221-1-(11)	*T. destructans*	OR961851	OR981297	OR973339	CAA
30	Guangdong	CSF25610	20230221-1-(12)	*T. destructans*	OR961852	OR981298	OR973340	AAA
30	Guangdong	CSF25611	20230221-1-(14)	*T. destructans*	OR961853	OR981299	OR973341	AAA
30	Guangdong	CSF25512	20230221-1-(24)	*T. epicoccoides*	OR961854	OR981300	OR973342	ACA
30	Guangdong	CSF25513 ^c^	20230221-1-(26)	*T. epicoccoides*	OR961855	OR981301	OR973343	AIA
30	Guangdong	CSF25514	20230221-1-(27)	*T. epicoccoides*	OR961856	OR981302	OR973344	ACA
30	Guangdong	CSF25515	20230221-1-(28)	*T. epicoccoides*	OR961857	OR981303	OR973345	ACA
30	Guangdong	CSF25516	20230221-1-(29)	*T. epicoccoides*	OR961858	OR981304	OR973346	AAA
30	Guangdong	CSF25517	20230221-1-(30)	*T. epicoccoides*	OR961859	OR981305	OR973347	ACA
31	Guangdong	CSF25587	20230220-3-(1)	*T. destructans*	OR961860	OR981306	OR973348	AAA
31	Guangdong	CSF25588	20230220-3-(2)	*T. destructans*	OR961861	OR981307	OR973349	AAA
31	Guangdong	CSF25589	20230220-3-(3)	*T. destructans*	OR961862	OR981308	OR973350	AAA
31	Guangdong	CSF25590	20230220-3-(4)	*T. destructans*	OR961863	OR981309	OR973351	AAA
31	Guangdong	CSF25591	20230220-3-(5)	*T. destructans*	OR961864	OR981310	OR973352	AAA
31	Guangdong	CSF25592	20230220-3-(6)	*T. destructans*	OR961865	OR981311	OR973353	AAA
31	Guangdong	CSF25593	20230220-3-(7)	*T. destructans*	OR961866	OR981312	OR973354	AAA
31	Guangdong	CSF25594	20230220-3-(8)	*T. destructans*	OR961867	OR981313	OR973355	AAA
31	Guangdong	CSF25595	20230220-3-(10)	*T. destructans*	OR961868	OR981314	OR973356	AAA
31	Guangdong	CSF25596	20230220-3-(11)	*T. destructans*	OR961869	OR981315	OR973357	AAA
31	Guangdong	CSF25597	20230220-3-(12)	*T. destructans*	OR961870	OR981316	OR973358	CAA
31	Guangdong	CSF25598	20230220-3-(13)	*T. destructans*	OR961871	OR981317	OR973359	CAA
31	Guangdong	CSF25500	20230220-3-(25)	*T. epicoccoides*	OR961872	OR981318	OR973360	ACA
31	Guangdong	CSF25501	20230220-3-(26)	*T. epicoccoides*	OR961873	OR981319	OR973361	ACA
31	Guangdong	CSF25502	20230220-3-(27)	*T. epicoccoides*	OR961874	OR981320	OR973362	AAA
31	Guangdong	CSF25503	20230220-3-(28)	*T. epicoccoides*	OR961875	OR981321	OR973363	ACA
31	Guangdong	CSF25504	20230220-3-(29)	*T. epicoccoides*	OR961876	OR981322	OR973364	ACA
31	Guangdong	CSF25505	20230220-3-(30)	*T. epicoccoides*	–	OR981323	OR973365	-EA
32	Guangdong	CSF25624	20230221-3-(3)	*T. destructans*	OR961877	OR981324	OR973366	CCA
32	Guangdong	CSF25625	20230221-3-(4)	*T. destructans*	OR961878	OR981325	OR973367	ACA
32	Guangdong	CSF25626	20230221-3-(5)	*T. destructans*	OR961879	OR981326	OR973368	AAA
32	Guangdong	CSF25627	20230221-3-(6)	*T. destructans*	OR961880	OR981327	OR973369	CAA
32	Guangdong	CSF25628	20230221-3-(7)	*T. destructans*	OR961881	OR981328	OR973370	AAA
32	Guangdong	CSF25629 ^c^	20230221-3-(8)	*T. destructans*	OR961882	OR981329	OR973371	ACA
32	Guangdong	CSF25630	20230221-3-(9)	*T. destructans*	OR961883	OR981330	OR973372	AAA
32	Guangdong	CSF25631	20230221-3-(10)	*T. destructans*	OR961884	OR981331	OR973373	AAA
32	Guangdong	CSF25632	20230221-3-(11)	*T. destructans*	OR961885	OR981332	OR973374	AAA
32	Guangdong	CSF25633	20230221-3-(12)	*T. destructans*	OR961886	OR981333	OR973375	AAA
32	Guangdong	CSF25634	20230221-3-(13)	*T. destructans*	OR961887	OR981334	OR973376	CCA
32	Guangdong	CSF25635	20230221-3-(14)	*T. destructans*	OR961888	OR981335	OR973377	CAA
32	Guangdong	CSF25636	20230221-3-(15)	*T. destructans*	OR961889	OR981336	OR973378	AAA
32	Guangdong	CSF25531	20230221-3-(18)	*T. epicoccoides*	OR961890	OR981337	OR973379	ACA
32	Guangdong	CSF25532	20230221-3-(19)	*T. epicoccoides*	–	OR981338	OR973380	-IA
32	Guangdong	CSF25533	20230221-3-(20)	*T. epicoccoides*	OR961891	OR981339	OR973381	ACA
32	Guangdong	CSF25534	20230221-3-(21)	*T. epicoccoides*	–	OR981340	OR973382	-AA
32	Guangdong	CSF25535	20230221-3-(22)	*T. epicoccoides*	OR961892	OR981341	OR973383	ACA
32	Guangdong	CSF25536	20230221-3-(23)	*T. epicoccoides*	–	OR981342	OR973384	-AA
33	Guangdong	CSF25576	20230220-2-(1)	*T. destructans*	OR961893	OR981343	OR973385	AAA
33	Guangdong	CSF25577	20230220-2-(2)	*T. destructans*	OR961894	OR981344	OR973386	AAA
33	Guangdong	CSF25578	20230220-2-(3)	*T. destructans*	OR961895	OR981345	OR973387	AAA
33	Guangdong	CSF25579	20230220-2-(4)	*T. destructans*	OR961896	OR981346	OR973388	AAA
33	Guangdong	CSF25580	20230220-2-(5)	*T. destructans*	OR961897	OR981347	OR973389	AAA
33	Guangdong	CSF25581	20230220-2-(6)	*T. destructans*	OR961898	OR981348	OR973390	CCA
33	Guangdong	CSF25582	20230220-2-(7)	*T. destructans*	OR961899	OR981349	OR973391	AAA
33	Guangdong	CSF25583	20230220-2-(8)	*T. destructans*	OR961900	OR981350	OR973392	CCA
33	Guangdong	CSF25584	20230220-2-(9)	*T. destructans*	OR961901	OR981351	OR973393	AAA
33	Guangdong	CSF25585	20230220-2-(11)	*T. destructans*	OR961902	OR981352	OR973394	AAA
33	Guangdong	CSF25586	20230220-2-(12)	*T. destructans*	OR961903	OR981353	OR973395	CAA
33	Guangdong	CSF25497	20230220-2-(18)	*T. epicoccoides*	OR961904	OR981354	OR973396	EAA
33	Guangdong	CSF25498	20230220-2-(19)	*T. epicoccoides*	OR961905	OR981355	OR973397	ECA
33	Guangdong	CSF25499	20230220-2-(20)	*T. epicoccoides*	OR961906	OR981356	OR973398	EAA
34	Guangdong	CSF25568	20230220-1-(2)	*T. destructans*	OR961907	OR981357	OR973399	AAA
34	Guangdong	CSF25569	20230220-1-(5)	*T. destructans*	OR961908	OR981358	OR973400	CCA
34	Guangdong	CSF25570	20230220-1-(7)	*T. destructans*	OR961909	OR981359	OR973401	CAA
34	Guangdong	CSF25571	20230220-1-(9)	*T. destructans*	OR961910	OR981360	OR973402	AAA
34	Guangdong	CSF25572	20230220-1-(10)	*T. destructans*	OR961911	OR981361	OR973403	CCA
34	Guangdong	CSF25573	20230220-1-(11)	*T. destructans*	OR961912	OR981362	OR973404	CAA
34	Guangdong	CSF25574	20230220-1-(14)	*T. destructans*	OR961913	OR981363	OR973405	AAA
34	Guangdong	CSF25575 ^c^	20230220-1-(16)	*T. destructans*	OR961914	OR981364	OR973406	AAA
34	Guangdong	CSF25495	20230220-1-(17)	*T. epicoccoides*	–	OR981365	OR973407	-EA
34	Guangdong	CSF25496	20230220-1-(18)	*T. epicoccoides*	–	OR981366	OR973408	-IA
35	Guangdong	CSF25371	20230216-5-(20)	*T. epicoccoides*	OR961915	OR981367	OR973409	CAA
35	Guangdong	CSF25372 ^c^	20230216-5-(22)	*T. epicoccoides*	OR961916	OR981368	OR973410	ECA
35	Guangdong	CSF25373	20230216-5-(24)	*T. epicoccoides*	OR961917	OR981369	OR973411	CAA
35	Guangdong	CSF25374 ^c^	20230216-5-(27)	*T. epicoccoides*	OR961918	OR981370	OR973412	EAA
35	Guangdong	CSF25375	20230216-5-(30)	*T. epicoccoides*	OR961919	OR981371	OR973413	EAA
35	Guangdong	CSF25376	20230216-5-(32)	*T. epicoccoides*	OR961920	OR981372	OR973414	ECA
36	Guangdong	CSF25356	20230216-3-(9)	*T. epicoccoides*	OR961921	OR981373	OR973415	AAA
36	Guangdong	CSF25357	20230216-3-(10)	*T. epicoccoides*	OR961922	OR981374	OR973416	EAA
36	Guangdong	CSF25358	20230216-3-(12)	*T. epicoccoides*	OR961923	OR981375	OR973417	ECA
36	Guangdong	CSF25359	20230216-3-(14)	*T. epicoccoides*	OR961924	OR981376	OR973418	EAA
36	Guangdong	CSF25360	20230216-3-(15)	*T. epicoccoides*	OR961925	OR981377	OR973419	CAA
36	Guangdong	CSF25361	20230216-3-(16)	*T. epicoccoides*	OR961926	OR981378	OR973420	ECA
37	Guangdong	CSF25367 ^c^	20230216-4-(1)	*T. epicoccoides*	OR961927	OR981379	OR973421	CCA
37	Guangdong	CSF25368 ^c^	20230216-4-(3)	*T. epicoccoides*	OR961928	OR981380	OR973422	BAA
37	Guangdong	CSF25369	20230216-4-(4)	*T. epicoccoides*	OR961929	OR981381	OR973423	EAA
37	Guangdong	CSF25370	20230216-4-(5)	*T. epicoccoides*	OR961930	OR981382	OR973424	EAA
38	Guangdong	CSF25347	20230216-2-(10)	*T. epicoccoides*	OR961931	OR981383	OR973425	EAA
38	Guangdong	CSF25348	20230216-2-(11)	*T. epicoccoides*	OR961932	OR981384	OR973426	AAA
38	Guangdong	CSF25349	20230216-2-(12)	*T. epicoccoides*	OR961933	OR981385	OR973427	CAA
38	Guangdong	CSF25350	20230216-2-(13)	*T. epicoccoides*	OR961934	OR981386	OR973428	AAA
38	Guangdong	CSF25351 ^c^	20230216-2-(14)	*T. epicoccoides*	OR961935	OR981387	OR973429	HAA
38	Guangdong	CSF25352	20230216-2-(15)	*T. epicoccoides*	OR961936	OR981388	OR973430	AAA
39	Guangdong	CSF25325	20230215-1-(1)	*T. epicoccoides*	OR961937	OR981389	OR973431	CCA
39	Guangdong	CSF25326 ^c^	20230215-1-(2)	*T. epicoccoides*	OR961938	OR981390	OR973432	CAA
39	Guangdong	CSF25327	20230215-1-(3)	*T. epicoccoides*	OR961939	OR981391	OR973433	EAA
39	Guangdong	CSF25328	20230215-1-(6)	*T. epicoccoides*	OR961940	OR981392	OR973434	ACA
39	Guangdong	CSF25329	20230215-1-(7)	*T. epicoccoides*	OR961941	OR981393	OR973435	EAA
39	Guangdong	CSF25330	20230215-1-(8)	*T. epicoccoides*	OR961942	OR981394	OR973436	EAA
40	Guangdong	CSF25336	20230216-1-(11)	*T. epicoccoides*	OR961943	OR981395	OR973437	ECA
40	Guangdong	CSF25337	20230216-1-(13)	*T. epicoccoides*	OR961944	OR981396	OR973438	ECA
40	Guangdong	CSF25338	20230216-1-(15)	*T. epicoccoides*	OR961945	OR981397	OR973439	EAA
40	Guangdong	CSF25339	20230216-1-(16)	*T. epicoccoides*	OR961946	OR981398	OR973440	AAA
40	Guangdong	CSF25340 ^c^	20230216-1-(17)	*T. epicoccoides*	OR961947	OR981399	OR973441	BCA
40	Guangdong	CSF25341	20230216-1-(18)	*T. epicoccoides*	OR961948	OR981400	OR973442	EAA
41	Guangdong	CSF25546	20230217-3-(1)	*T. destructans*	OR961949	OR981401	OR973443	CCA
41	Guangdong	CSF25547 ^c^	20230217-3-(2)	*T. epicoccoides*	OR961950	OR981402	OR973444	XBA
41	Guangdong	CSF25548	20230217-3-(6)	*T. destructans*	OR961951	OR981403	OR973445	AAA
41	Guangdong	CSF25549	20230217-3-(8)	*T. destructans*	OR961952	OR981404	OR973446	CAA
41	Guangdong	CSF25550	20230217-3-(13)	*T. destructans*	OR961953	OR981405	OR973447	AAA
41	Guangdong	CSF25551	20230217-3-(16)	*T. destructans*	OR961954	OR981406	OR973448	AAA
41	Guangdong	CSF25552	20230217-3-(17)	*T. destructans*	OR961955	OR981407	OR973449	AAA
41	Guangdong	CSF25553	20230217-3-(21)	*T. destructans*	OR961956	OR981408	OR973450	CAA
41	Guangdong	CSF25554	20230217-3-(24)	*T. destructans*	OR961957	OR981409	OR973451	CCA
41	Guangdong	CSF25555	20230217-3-(25)	*T. destructans*	OR961958	OR981410	OR973452	CAA
41	Guangdong	CSF25556	20230217-3-(30)	*T. destructans*	OR961959	OR981411	OR973453	CAA
41	Guangdong	CSF25557 ^c^	20230217-3-(37)	*T. destructans*	OR961960	OR981412	OR973454	CAA
41	Guangdong	CSF25399	20230217-3-(51)	*T. epicoccoides*	OR961961	OR981413	OR973455	AAA
41	Guangdong	CSF25400	20230217-3-(52)	*T. epicoccoides*	OR961962	OR981414	OR973456	AAA
41	Guangdong	CSF25401	20230217-3-(53)	*T. epicoccoides*	OR961963	OR981415	OR973457	CAA
41	Guangdong	CSF25402	20230217-3-(54)	*T. epicoccoides*	OR961964	OR981416	OR973458	ECA
41	Guangdong	CSF25403	20230217-3-(55)	*T. epicoccoides*	OR961965	OR981417	OR973459	AAA
41	Guangdong	CSF25404	20230217-3-(56)	*T. epicoccoides*	OR961966	OR981418	OR973460	ECA
42	Guangdong	CSF25558	20230219-1-(5)	*T. destructans*	OR961967	OR981419	OR973461	AAA
42	Guangdong	CSF25559	20230219-1-(8)	*T. destructans*	OR961968	OR981420	OR973462	AAA
42	Guangdong	CSF25560	20230219-1-(9)	*T. destructans*	OR961969	OR981421	OR973463	CAA
42	Guangdong	CSF25561	20230219-1-(15)	*T. destructans*	OR961970	OR981422	OR973464	AAA
42	Guangdong	CSF25562	20230219-1-(17)	*T. destructans*	OR961971	OR981423	OR973465	AAA
42	Guangdong	CSF25563	20230219-1-(23)	*T. destructans*	OR961972	OR981424	OR973466	CAA
42	Guangdong	CSF25564	20230219-1-(25)	*T. destructans*	OR961973	OR981425	OR973467	CAA
42	Guangdong	CSF25565	20230219-1-(26)	*T. destructans*	OR961974	OR981426	OR973468	CAA
42	Guangdong	CSF25566	20230219-1-(27)	*T. destructans*	OR961975	OR981427	OR973469	AAA
42	Guangdong	CSF25567	20230219-1-(29)	*T. destructans*	OR961976	OR981428	OR973470	CAA
43	Guangdong	CSF25390	20230217-2-(18)	*T. epicoccoides*	OR961977	OR981429	OR973471	ECA
43	Guangdong	CSF25391	20230217-2-(20)	*T. epicoccoides*	OR961978	OR981430	OR973472	AAA
43	Guangdong	CSF25392	20230217-2-(21)	*T. epicoccoides*	OR961979	OR981431	OR973473	AAA
43	Guangdong	CSF25393	20230217-2-(24)	*T. epicoccoides*	OR961980	OR981432	OR973474	EAA
43	Guangdong	CSF25394	20230217-2-(25)	*T. epicoccoides*	OR961981	OR981433	OR973475	EAA
43	Guangdong	CSF25395	20230217-2-(26)	*T. epicoccoides*	OR961982	OR981434	OR973476	EAA
44	Guangdong	CSF25378	20230217-1-(13)	*T. epicoccoides*	OR961983	OR981435	OR973477	ECA
44	Guangdong	CSF25379	20230217-1-(14)	*T. epicoccoides*	OR961984	OR981436	OR973478	CAA
44	Guangdong	CSF25380	20230217-1-(15)	*T. epicoccoides*	OR961985	OR981437	OR973479	ECA
44	Guangdong	CSF25381	20230217-1-(17)	*T. epicoccoides*	OR961986	OR981438	OR973480	EAA
44	Guangdong	CSF25382	20230217-1-(18)	*T. epicoccoides*	OR961987	OR981439	OR973481	EAA
44	Guangdong	CSF25383	20230217-1-(21)	*T. epicoccoides*	OR961988	OR981440	OR973482	EAA
45	Guangdong	CSF25436 ^c^	20230218-1-(6)	*T. epicoccoides*	OR961989	OR981441	OR973483	ACA
45	Guangdong	CSF25437	20230218-1-(7)	*T. epicoccoides*	OR961990	OR981442	OR973484	EAA
45	Guangdong	CSF25438	20230218-1-(8)	*T. epicoccoides*	OR961991	OR981443	OR973485	AAA
45	Guangdong	CSF25439	20230218-1-(9)	*T. epicoccoides*	OR961992	OR981444	OR973486	EAA
45	Guangdong	CSF25440	20230218-1-(10)	*T. epicoccoides*	OR961993	OR981445	OR973487	EAA
45	Guangdong	CSF25441	20230218-1-(11)	*T. epicoccoides*	OR961994	OR981446	OR973488	BAA
46	Guangdong	CSF25419	20230217-4-(3)	*T. epicoccoides*	OR961995	OR981447	OR973489	EAA
46	Guangdong	CSF25420	20230217-4-(4)	*T. epicoccoides*	OR961996	OR981448	OR973490	AAA
46	Guangdong	CSF25421	20230217-4-(5)	*T. epicoccoides*	OR961997	OR981449	OR973491	AAA
46	Guangdong	CSF25422	20230217-4-(6)	*T. epicoccoides*	OR961998	OR981450	OR973492	ECA
46	Guangdong	CSF25423	20230217-4-(7)	*T. epicoccoides*	–	OR981451	OR973493	-CA
46	Guangdong	CSF25424	20230217-4-(8)	*T. epicoccoides*	OR961999	OR981452	OR973494	EAA
47	Guangdong	CSF25451	20230218-2-(7)	*T. epicoccoides*	OR962000	OR981453	OR973495	AAA
47	Guangdong	CSF25452	20230218-2-(8)	*T. epicoccoides*	OR962001	OR981454	OR973496	EAA
47	Guangdong	CSF25453	20230218-2-(9)	*T. epicoccoides*	OR962002	OR981455	OR973497	BAA
47	Guangdong	CSF25454	20230218-2-(10)	*T. epicoccoides*	OR962003	OR981456	OR973498	AAA
47	Guangdong	CSF25455	20230218-2-(11)	*T. epicoccoides*	OR962004	OR981457	OR973499	ECA
47	Guangdong	CSF25456	20230218-2-(13)	*T. epicoccoides*	OR962005	OR981458	OR973500	ACA
48	Guangdong	CSF25468	20230218-3-(12)	*T. epicoccoides*	OR962006	OR981459	OR973501	AAA
48	Guangdong	CSF25469	20230218-3-(13)	*T. epicoccoides*	OR962007	OR981460	OR973502	ACA
48	Guangdong	CSF25470	20230218-3-(14)	*T. epicoccoides*	OR962008	OR981461	OR973503	AAA
48	Guangdong	CSF25471	20230218-3-(15)	*T. epicoccoides*	OR962009	OR981462	OR973504	ACA
48	Guangdong	CSF25472	20230218-3-(16)	*T. epicoccoides*	OR962010	OR981463	OR973505	EAA
48	Guangdong	CSF25473	20230218-3-(17)	*T. epicoccoides*	OR962011	OR981464	OR973506	EAA
49	Guangdong	CSF25482	20230218-4-(2)	*T. epicoccoides*	OR962012	OR981465	OR973507	AAA
49	Guangdong	CSF25483	20230218-4-(4)	*T. epicoccoides*	OR962013	OR981466	OR973508	EAA
49	Guangdong	CSF25484	20230218-4-(5)	*T. epicoccoides*	OR962014	OR981467	OR973509	BAA
49	Guangdong	CSF25485	20230218-4-(6)	*T. epicoccoides*	OR962015	OR981468	OR973510	EAA
49	Guangdong	CSF25486	20230218-4-(7)	*T. epicoccoides*	OR962016	OR981469	OR973511	ACA
49	Guangdong	CSF25487	20230218-4-(8)	*T. epicoccoides*	OR962017	OR981470	OR973512	EAA
50	Fujian	CSF25801	20230330-1-(21)	*T. epicoccoides*	OR962018	OR981471	OR973513	ACA
50	Fujian	CSF25802	20230330-1-(22)	*T. epicoccoides*	OR962019	OR981472	OR973514	EAA
50	Fujian	CSF25803	20230330-1-(23)	*T. epicoccoides*	OR962020	OR981473	OR973515	AAA
50	Fujian	CSF25804	20230330-1-(24)	*T. epicoccoides*	OR962021	OR981474	OR973516	AAA
50	Fujian	CSF25805	20230330-1-(25)	*T. epicoccoides*	OR962022	OR981475	OR973517	ECA
51	Fujian	CSF25773 ^c^	20230328-1-(2)	*T. epicoccoides*	OR962023	OR981476	OR973518	DIA
51	Fujian	CSF25774	20230328-1-(5)	*T. epicoccoides*	OR962024	OR981477	OR973519	DIA
51	Fujian	CSF25775	20230328-1-(7)	*T. epicoccoides*	OR962025	OR981478	OR973520	DIA
51	Fujian	CSF25776	20230328-1-(8)	*T. epicoccoides*	OR962026	OR981479	OR973521	DIA
51	Fujian	CSF25777	20230328-1-(9)	*T. epicoccoides*	OR962027	OR981480	OR973522	ECA
51	Fujian	CSF25778	20230328-1-(11)	*T. epicoccoides*	OR962028	OR981481	OR973523	AAA
52	Fujian	CSF25782	20230329-1-(1)	*T. epicoccoides*	–	OR981482	OR973524	-EA
52	Fujian	CSF25783	20230329-1-(2)	*T. epicoccoides*	OR962029	OR981483	OR973525	EAA
52	Fujian	CSF25784	20230329-1-(3)	*T. epicoccoides*	OR962030	OR981484	OR973526	EAA
52	Fujian	CSF25785	20230329-1-(4)	*T. epicoccoides*	OR962031	OR981485	OR973527	ACA
52	Fujian	CSF25786	20230329-1-(5)	*T. epicoccoides*	–	OR981486	OR973528	-CA
52	Fujian	CSF25787	20230329-1-(6)	*T. epicoccoides*	OR962032	OR981487	OR973529	EAA
53	Fujian	CSF25790	20230329-2-(1)	*T. epicoccoides*	OR962033	OR981488	OR973530	DIA
53	Fujian	CSF25791	20230329-2-(4)	*T. epicoccoides*	OR962034	OR981489	OR973531	DIA
53	Fujian	CSF25792	20230329-2-(5)	*T. epicoccoides*	OR962035	OR981490	OR973532	DIA
53	Fujian	CSF25793	20230329-2-(6)	*T. epicoccoides*	OR962036	OR981491	OR973533	DIA
53	Fujian	CSF25794	20230329-2-(7)	*T. epicoccoides*	–	OR981492	OR973534	-IA
53	Fujian	CSF25795	20230329-2-(8)	*T. epicoccoides*	OR962037	OR981493	OR973535	DIA
54	Fujian	CSF25850	20230331-3-(1)	*T. epicoccoides*	OR962038	OR981494	OR973536	CAA
54	Fujian	CSF25851	20230331-3-(2)	*T. epicoccoides*	OR962039	OR981495	OR973537	ECA
54	Fujian	CSF25852	20230331-3-(4)	*T. epicoccoides*	OR962040	OR981496	OR973538	EAA
54	Fujian	CSF25853	20230331-3-(5)	*T. epicoccoides*	OR962041	OR981497	OR973539	CAA
54	Fujian	CSF25854	20230331-3-(6)	*T. epicoccoides*	OR962042	OR981498	OR973540	CAA
54	Fujian	CSF25855	20230331-3-(7)	*T. epicoccoides*	OR962043	OR981499	OR973541	EAA
55	Fujian	CSF25840 ^c^	20230331-2-(2)	*T. epicoccoides*	OR962044	OR981500	OR973542	BAA
55	Fujian	CSF25841	20230331-2-(3)	*T. epicoccoides*	OR962045	OR981501	OR973543	ECA
55	Fujian	CSF25842	20230331-2-(4)	*T. epicoccoides*	OR962046	OR981502	OR973544	EAA
55	Fujian	CSF25843	20230331-2-(5)	*T. epicoccoides*	OR962047	OR981503	OR973545	AAA
55	Fujian	CSF25844	20230331-2-(6)	*T. epicoccoides*	OR962048	OR981504	OR973546	AAA
55	Fujian	CSF25845	20230331-2-(9)	*T. epicoccoides*	OR962049	OR981505	OR973547	EAA
56	Fujian	CSF25830	20230331-1-(1)	*T. epicoccoides*	OR962050	OR981506	OR973548	ACA
56	Fujian	CSF25831	20230331-1-(2)	*T. epicoccoides*	OR962051	OR981507	OR973549	EAA
56	Fujian	CSF25832	20230331-1-(3)	*T. epicoccoides*	OR962052	OR981508	OR973550	BAA
56	Fujian	CSF25833	20230331-1-(5)	*T. epicoccoides*	OR962053	OR981509	OR973551	CCA
56	Fujian	CSF25834	20230331-1-(6)	*T. epicoccoides*	OR962054	OR981510	OR973552	EAA
56	Fujian	CSF25835	20230331-1-(9)	*T. epicoccoides*	OR962055	OR981511	OR973553	AAA
57	Fujian	CSF25820	20230330-3-(1)	*T. epicoccoides*	OR962056	OR981512	OR973554	EAA
57	Fujian	CSF25821	20230330-3-(2)	*T. epicoccoides*	OR962057	OR981513	OR973555	ECA
57	Fujian	CSF25822	20230330-3-(3)	*T. epicoccoides*	OR962058	OR981514	OR973556	CAA
57	Fujian	CSF25823	20230330-3-(4)	*T. epicoccoides*	OR962059	OR981515	OR973557	ECA
57	Fujian	CSF25824	20230330-3-(5)	*T. epicoccoides*	OR962060	OR981516	OR973558	EAA
57	Fujian	CSF25825	20230330-3-(6)	*T. epicoccoides*	OR962061	OR981517	OR973559	ECA
58	Fujian	CSF25810	20230330-2-(1)	*T. epicoccoides*	OR962062	OR981518	OR973560	AAA
58	Fujian	CSF25811	20230330-2-(3)	*T. epicoccoides*	OR962063	OR981519	OR973561	CCA
58	Fujian	CSF25812	20230330-2-(4)	*T. epicoccoides*	OR962064	OR981520	OR973562	CAA
58	Fujian	CSF25813	20230330-2-(5)	*T. epicoccoides*	OR962065	OR981521	OR973563	CCA
58	Fujian	CSF25814	20230330-2-(6)	*T. epicoccoides*	OR962066	OR981522	OR973564	EAA
58	Fujian	CSF25815	20230330-2-(8)	*T. epicoccoides*	OR962067	OR981523	OR973565	ACA
59	Fujian	CSF25860	20230401-1-(1)	*T. epicoccoides*	OR962068	OR981524	OR973566	AAA
59	Fujian	CSF25861	20230401-1-(2)	*T. epicoccoides*	OR962069	OR981525	OR973567	EAA
59	Fujian	CSF25862	20230401-1-(3)	*T. epicoccoides*	OR962070	OR981526	OR973568	AAA
59	Fujian	CSF25863	20230401-1-(4)	*T. epicoccoides*	OR962071	OR981527	OR973569	EAA
59	Fujian	CSF25864	20230401-1-(5)	*T. epicoccoides*	OR962072	OR981528	OR973570	EAA
59	Fujian	CSF25865	20230401-1-(7)	*T. epicoccoides*	OR962073	OR981529	OR973571	EAA

^a^ Code of 59 sampling sites connecting to “Sampling Site No.” in Table 1. ^b^ CSF: Culture Collection located at Research Institute of Fasting-growing Trees (RIFT), Chinese Academy of Forestry, Zhanjiang, Guangdong Province, China. ^c^ Isolates used for phylogenetic analyses in this study. ^d^ Information associated with sample site and isolate, for example, “20230315-2-(2)” indicates sample number “20230315-2-(2)” and isolate from this sample; the sample number connecting to “Sample and Isolate Information” in Table 1. ^e^ ITS = internal transcribed spacer regions and intervening 5.8S nrRNA gene; *tef1* = translation elongation factor 1-alpha; *tub2* = β-tubulin. ^f^ “–” represents the relative locus that was not successfully amplified in this study. ^g^ Genotype within each *Teratosphaeria* species, determined by sequences of the ITS, *tef1* and *tub2* regions. The same letter among isolates from each species means they shared the same genotype; “-” means not available.

Sequences downloaded from the NCBI database and sequences generated in this study were aligned using MAFFT online v. 7 (http://mafft.cbrc.jp/alignment/server/ (accessed on 2 November 2023)) [50], with the iterative refinement method (FFT-NS-i setting). The alignments were further edited manually with MEGA v. 7.0 software [49] when necessary. Sequences of each of the ITS, *tef1*, and *tub2* gene regions, as well as the combination of these three gene regions, were analyzed.

Maximum likelihood (ML) analyses were conducted for the three individual gene sequences of ITS, *tef1*, and *tub2*, as well as for a concatenated dataset of all three genes. ML analyses were performed with RaxML v. 8.2.4 on the CIPRES Science Gateway v. 3.3 [51], with default GTR substitution matrix and 1000 bootstrap replicates [52]. Phylogenetic trees were viewed using MEGA v. 7.0 [49]. Sequence data of one isolate of *Staninwardia suttonii* (CBS 120061) were used as the outgroup [30].

## 3. Results

### 3.1. Fungal Isolations

Diseased leaves with conidiomata of *Mycosphaerellaceae* and *Teratosphaeriaceae* were collected from 59 sites in five provinces in southern China (Table 1; Figure 2). Single conidia from the conidiomata on the diseased leaves were transferred to fresh 2% MEA for fungal isolation. These conidia exhibited the typical morphological characteristics of *Teratosphaeria* species. Two groups of fungi corresponding to the disease symptoms were observed. The conidia of the first group of fungi were generally brown, and straight to slightly curved in shape. The number of septa in the conidia ranged from one to seven; the majority of conidia had three to five septa (Figure 3). The conidia of the second group of fungi were light brown, variously curved, and rarely straight. The conidia had one to three septa; the majority of conidia had three septa (Figure 4). The conidia of both groups of fungi were base truncate and apex obtuse. The conidia of the first group of fungi were wider, straighter, and darker than those of the second group (Figure 3 and Figure 4). These two groups of fungi had the typical morphological characteristics of *T. epicoccoides* and *T. destructans*, respectively [43,53]. In total, 558 isolates were obtained from the 59 sites. Three to twenty isolates were obtained from each sampled site, depending on the number of samples collected and the morphological variations in the conidia on the diseased leaves (Table 1).

Both groups of fungi were isolated from plantations of *E. urophylla*, *E. urophylla* × *E. grandis*, *E. urophylla* × *E. pellita* and *E. urophylla* × *E. tereticomis*, and most especially from multiple genotypes of *E. urophylla* × *E. grandis*, which were widely planted in the sampled regions. In addition, the first group of fungi were also frequently observed on the majority of other *Eucalyptus* genotypes in the plantations, including species of *E. camaldulensis*, *E. grandis*, *E. pellita* and their hybrids with *E. urophylla*. *T. destructans* is limited to certain species of *E. grandis*, *E. urophylla*, *E. tereticornis* and their hybrids.

### 3.2. Multi-Gene Phylogenetic Analyses and Species Identification

The BLAST results indicated that the 558 isolates obtained and sequenced in this study were mainly categorized into two groups. These two groups of fungi were similar to *T. epicoccoides* (Group A) and *T. destructans* (Group B), respectively. Amplicons generated for the ITS, *tef1*, and *tub2* gene regions of fungi similar to *T. epicoccoides* were approximately 500, 350, and 350 bp, respectively. The sequences of ITS, *tef1*, and *tub2* gene regions of fungi similar to *T. destructans* were approximately 290, 250, and 350 bp, respectively. Sequences of two isolates for each genotype generated by the three genes were used in the analyses. Sequences of ex-type specimen strains and other strains of 29 *Teratosphaeria* species, including *T. epicoccoides* and *T. destructans*, closely related to isolates obtained in the current study, were downloaded from GenBank for sequence comparisons and phylogenetic analyses (Table 3).

Based on the phylogenetic analyses of the ITS, *tef1*, *tub2*, and the combined datasets, the isolates of two groups of fungi, Group A and Group B in this study, were most closely related to *T. epicoccoides* and *T. destructans*, respectively. The isolates of Group A were consistently grouped with, or close to, the ex-type isolate CMW 5348 of *T. epicoccoides* in each of the ITS, *tef1*, *tub2*, and the combined dataset trees (Figure 5, Figure 6, Figure 7 and Figure 8). Additionally, some isolates of Group A formed independent clades in each of the four phylogenetic trees (Figure 5, Figure 6, Figure 7 and Figure 8). This is consistent with the phylogenetic analysis results presented by Taole et al. [43]. Analysis of the sequence data of the three regions resulted in incongruent genealogies. There was no evidence of distinct species boundaries. Isolates of Group A were identified as *T. epicoccoides*.

In Group B, all isolates were grouped with ex-type isolate CBS 111370 of *T. destructans* in the *tub2* tree (Figure 8). Additionally, some isolates of Group B formed independent clades in each of the ITS and *tef1* phylogenetic trees. However, these isolates did not form consistently independent clades, nor were they supported by high bootstrap values in both the ITS and *tef1* phylogenetic trees (Figure 6 and Figure 7). For example, isolates of genotypes CAA and CBA were grouped together and formed one independent clade with high bootstrap value (89%) in the ITS tree (Figure 6). These isolates were grouped with ex-type isolates CBS 111369 and CBS 111370 of *T. destructans* in the *tef1* tree (Figure 7). The results suggest that these reflect intraspecific sequence differences rather than interspecies variation. Combining the phylogenetic analysis results of ITS, *tef1*, *tub2* and the datasets (Figure 5, Figure 6, Figure 7 and Figure 8), the isolates in Group B were identified as *T. destructans*.

### 3.3. Distribution of Teratosphaeria Species

Based on the DNA sequence comparisons of ITS, *tef1*, and *tub2* sequences, the 558 isolates obtained from 59 sampling sites in this study were identified as *T. epicoccoides* (312 isolates; 55.9%) and *T. destructans* (246 isolates, 44.1%). *T. epicoccoides* was isolated and identified in 56 (accounting 95%) sampling sites, and *T. destructans* was obtained from 27 (accounting 46%) sampling sites. These two species were all obtained from 24 (accounting for 41%) sampling sites (Figure 2).

Determined by ITS, *tef1*, and *tub2* sequences, 21 genotypes were generated for the 291 *T. epicoccoides* isolates (the genotypes of 21 isolates were not clear because the sequences of three gene regions were not all obtained) (Table 4). Six genotypes were generated for the 242 *T. destructans* isolates (the genotypes of four isolates were not clear because the sequences of three gene regions were not all obtained) (Table 5). The ratios of the genotype number to the isolate number of *T. epicoccoides* and *T. destructans* were all highest in Hainan Province.

## 4. Discussion

In this study, systematical disease surveys were conducted to collect diseased *Eucalyptus* leaves with typical fruiting structures of *Mycosphaerellaceae* and *Teratosphaeriaceae* in the core *Eucalyptus* plantation regions in China. In total, 558 isolates were identified according to disease symptoms and morphological characteristics and were mainly based on DNA sequence comparisons of three gene regions. These fungi were identified as *T. epicoccoides* and *T. destructans*. The results of this study indicate that *T. epicoccoides* and *T. destructans* are widely distributed in the core *Eucalyptus* planting regions in southern China.

This study has clarified and expanded the geographic distribution of *Teratosphaeria* species associated with diseased *Eucalyptus* leaves in southern China. *Teratosphaeria epicoccoides* was initially described in 1992. It was indicated that this species is distributed in many countries and regions worldwide. The leaf specimens used for the initial description included a total of 25 species of *Eucalyptus* and *Corymbia* in Australia, Brazil, Argentina, Zambia, India, and Ethiopia [54]. Currently, *T. epicoccoides* is widely reported on *Eucalyptus* leaves globally [41,43]. *Teratosphaeria destructans* was initially reported and described in 1996 on the leaves of *E. grandis* in Indonesia, along with some other unknown *Eucalyptus* species [53]. Currently, *T. destructans* is reported in a global range, including countries in Africa such as South Africa, and countries in Asia such as China, Thailand, and East Timor [41,55]. In China, *T. epicoccoides* and *T. destructans* were both first isolated from leaves of *E. urophylla* in Guangdong Province in 2006 [13]. Prior to this study, the distribution reports of these two species in China were both limited to Guangdong Province [13,26]. The results of this study suggest that *T. epicoccoides* is distributed in areas that are majority-planted with *Eucalyptus* in southern China, and *T. destructans* is also widely distributed throughout large regions in this country.

DNA sequence comparisons were considered a reliable method for the classification and identification of *Mycosphaerellaceae* and *Teratosphaeriaceae* [23,30]. For species of *Mycosphaerellaceae* and *Teratosphaeriaceae*, traditional classification has primarily relied on morphological characteristics. When classifying based on morphological characteristics, the identification of species of *Mycosphaerellaceae* and *Teratosphaeriaceae* is influenced by the conserved morphological features of their respective sexual morphs [56,57,58]. Scientists have therefore shifted the focus of classifying and identifying these species mostly towards their asexual morphs [59,60,61]. However, similar asexual morphologies have also independently evolved from different taxa, adding further complexity to the taxonomy of these species [62]. In 2014, Quaedvlieg et al. proposed that the accurate classification of genera and species within *Mycosphaerellaceae* and *Teratosphaeriaceae* cannot be achieved solely through morphological comparisons. Instead, it requires the integration of molecular data [30]. The research results indicated that the ITS gene serves as a primary barcode locus to distinguish taxa in *Mycosphaerella* and *Teratosphaeria* associated with *Eucalyptus* leaf disease. The ITS gene is easily generated and has the most extensive dataset available; therefore, *tef1*, *tub*, or rpb2 represent useful secondary barcode loci [30]. In this study, species identification was primarily based on the comparison of three gene sequences of ITS, *tef1*, and *tub2.* These three gene sequences have been widely employed to differentiate both intra- and inter-specific variations among species in *Teratosphaeriaceae* [25,43,55].

To date, eight species of *Mycosphaerellaceae* and *Teratosphaeriaceae* isolated from *Eucalyptus* leaves in China have been identified based on DNA sequence comparisons. These species include *Neoceratosperma yunnanensis*, *Pallidocercospora crystallina*, *Paramycosphaerella marksii*, *Pseudocercospora flavomarginata*, *Pse*. *gracilis*, *Pse*. *haiweiensis*, *Teratosphaeria destructans*, and *T. epicoccoides* [26,27,29,30,63]. With the exception of two species of *Teratosphaeria*, the other six species reside in *Mycosphaerellaceae*. It is still unknown whether these eight species are all pathogenic to *Eucalyptus* in China because no pathogenicity tests have been conducted.

One shortcoming of this study is that the pathogenicity of the two identified *Teratosphaeria* species was not investigated. At present, on a global scale, methods for assessing the pathogenicity of *Mycosphaerellaceae* and *Teratosphaeriaceae* species isolated from diseased *Eucalyptus* leaves include spraying ascospore suspensions, conidial suspensions, hyphal fragment suspensions, or a mixture of spores and hyphal fragment suspensions onto *Eucalyptus* seedlings in controlled environments. Additional inoculation methods involve applying ascospore masses or lesions from diseased leaf areas onto *Eucalyptus* seedling leaves [44,57,64]. Conidial suspensions are considered to be a typical method for evaluating the pathogenicity of *Mycosphaerellaceae* and *Teratosphaeriaceae* species; however, this is not possible for species that do not sporulate sufficiently in culture. Very limited research has been conducted on the pathogenicity of *T. destructans* and *T. epicoccoides* on *Eucalyptus*. The *Teratosphaeria* isolates obtained in the current study failed to produce conidia on media. In order to test the pathogenicity of *T. destructans* and *T. epicoccoides*, it is vital to find a way to sufficiently induce the sporulation of these fungi in culture or find a new method to test their pathogenicity. For example, the hyphal fragment suspensions were replaced with conidial suspensions to test the pathogenicity of *Calonectria pseudoreteaudii* [65]. Hyphal fragment suspension may also be used to assess the pathogenicity of *Teratosphaeria* species, although this need to be evaluated. Previous research results indicated that *T. epicoccoides* should be indigenous to eastern Australia, and spread to western Australia and other regions of the world [41,66]. *T. destructans* is recognized as absent from Australia. The origin of *T. destructans* remains unknown, but it is likely to have originated in Indonesia or East Timor [41,67,68,69]. Research results support the hypothesis that *T. epicoccoides* and *T. destructans* are spread via the human-mediated movement of infected plants and seeds [41,66,70]. This suggests the need for much more stringent and more sophisticated biosecurity measures, such as the use of metabarcoding approaches to test seedlots and plant materials under quarantine [41,67,68].

The results of this study confirmed that both *T. epicoccoides* and *T. destructans* are dominant species on diseased *Eucalyptus* leaves in China. Both species are widely distributed in *Eucalyptus* plantations that are mainly distributed across regions of southern China. Although previous studies have indicated that *T. epicoccoides* and *T. destructans* are widely distributed on *Eucalyptus* leaves globally [41,44], research on these species on diseased *Eucalyptus* leaves specifically in China is limited. During the sampling process in the current study, we found that the severity caused by each species of *T. epicoccoides* and *T. destructans* on different genotypes of *E. urophylla* × *E. grandis* were different. Research results showed that significant differences in resistance exist among the six tested *E. grandis* × *E. urophylla* genotypes to the inoculated *T. destructans* [71]. It is suggested that breeding and selection of *Eucalyptus* tolerant/resistant species, hybrids and clones are needed for the future management of leaf diseases caused by *Teratosphaeria* species [41]. Future research should focus on exploring pathogenicity testing methods for *T. epicoccoides* and *T. destructans* and clarifying their pathogenic characteristics. This will guide the selection of disease-resistant *Eucalyptus* genotypes for these two widely distributed species.

## 5. Conclusions

This study has proven that both *T. epicoccoides* and *T. destructans* are dominant species and widely distributed on diseased *Eucalyptus* leaves in southern China; both *T. epicoccoides* and *T. destructans* were present in each of the five sampled provinces with multiple genotypes. These two species are not clonal in China. The genetic diversities of *T. epicoccoides* and *T. destructans* were high in the southern sampled province. The wider distribution, and higher genetic diversity than has previously been considered indicates that both *T. epicoccoides* and *T. destructans* should be considered seriously in future disease management schemes.

## Figures and Tables

**Figure 1 microorganisms-12-00129-f001:**
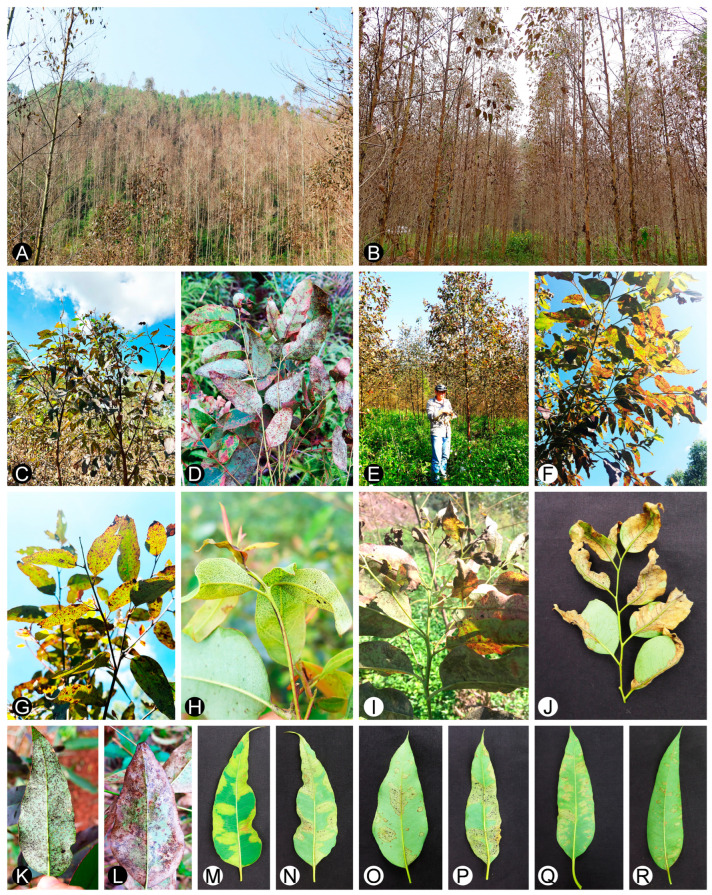
Disease symptoms associated with *Teratosphaeria epicoccoides* and *T. destructans* on *Eucalyptus* plantations in southern China: (**A**,**B**) trees in 2-year-old *Eucalyptus urophylla* × *E. grandis* plantations associated with *T. epicoccoides* and *T. destructans*. Leaves of the whole trees were infected and resulted in intense defoliation in the tree growth season; (**C**,**D**) typical leaf spot, vein delimitation, and chlorosis symptoms on leaves of *E. urophylla* × *E. grandis* associated with *T. epicoccoides*. New leaves on the top of shoots emerged after infection (**C**); (**E**,**F**) all leaves of 0.5-year-old *E. urophylla* × *E. grandis* trees in one plantation infected by *T. destructans*; (**G**,**H**) typical disease symptoms on *E. urophylla* × *E. grandis* leaves caused by *T. epicoccoides* (**G**) and *T. destructans* (**H**); (**I**,**J**) leaf blighted after infection by *T. destructans*; (**K**,**L**) heavy sporulation of *T. epicoccoides*; (**M**,**N**) water-soaked and chlorosis symptoms caused by *T. destructans* on the adaxial (**M**) and abaxial (**N**) leaf surface; and (**O**–**R**) Four *E. urophylla* × *E. grandis* hybrid genotypes exhibiting chlorosis and leaf spot caused by *T. destructans*.

**Figure 2 microorganisms-12-00129-f002:**
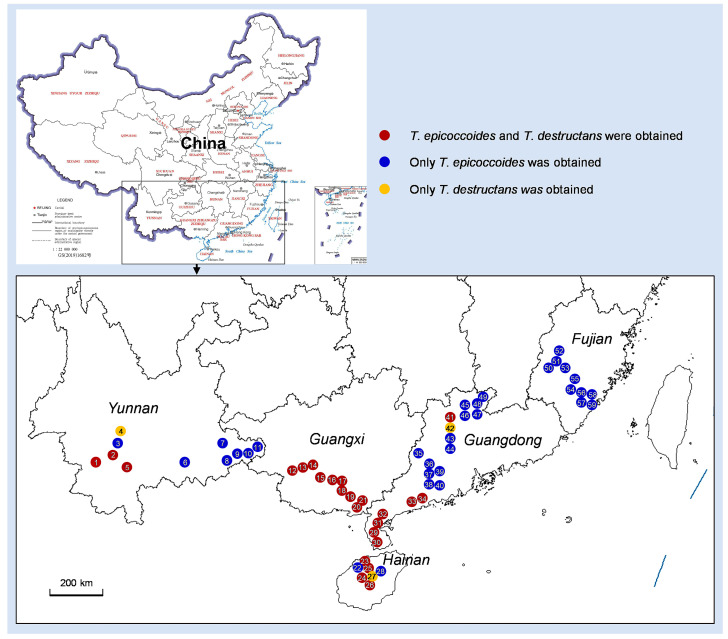
Map showing the 59 sites in the five provinces in China where the diseased leaf samples were collected, and the *Teratosphaeria* species identified in each site.

**Figure 3 microorganisms-12-00129-f003:**
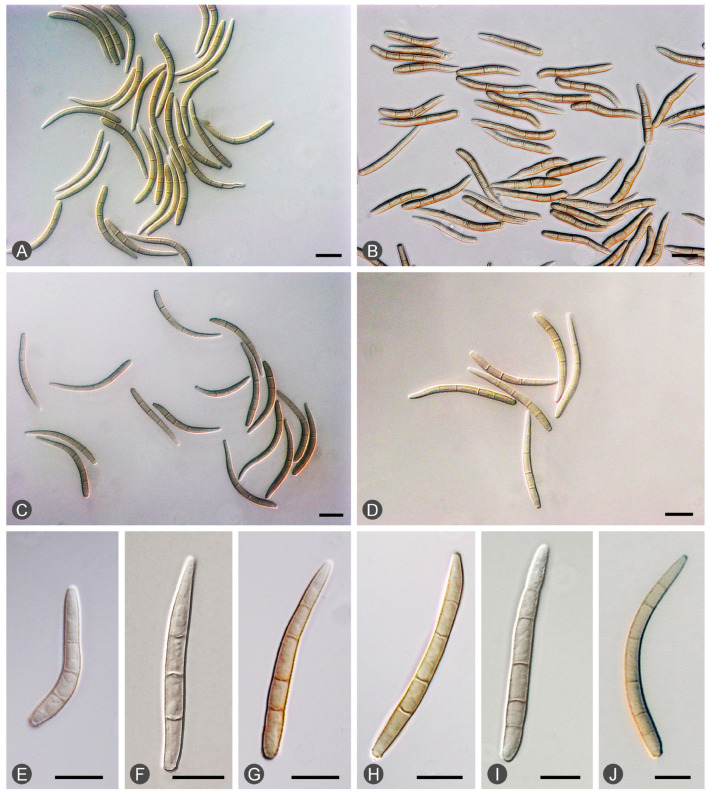
Morphological features of *Teratosphaeria epicoccoides*: (**A**–**D**) slightly curved conidia with septa from different *Eucalyptus* trees. (**E**–**J**) Conidia with three to six septa: three septa (**E**,**F**); four septa (**G**,**H**); five septa (**I**); and six septa (**J**).

**Figure 4 microorganisms-12-00129-f004:**
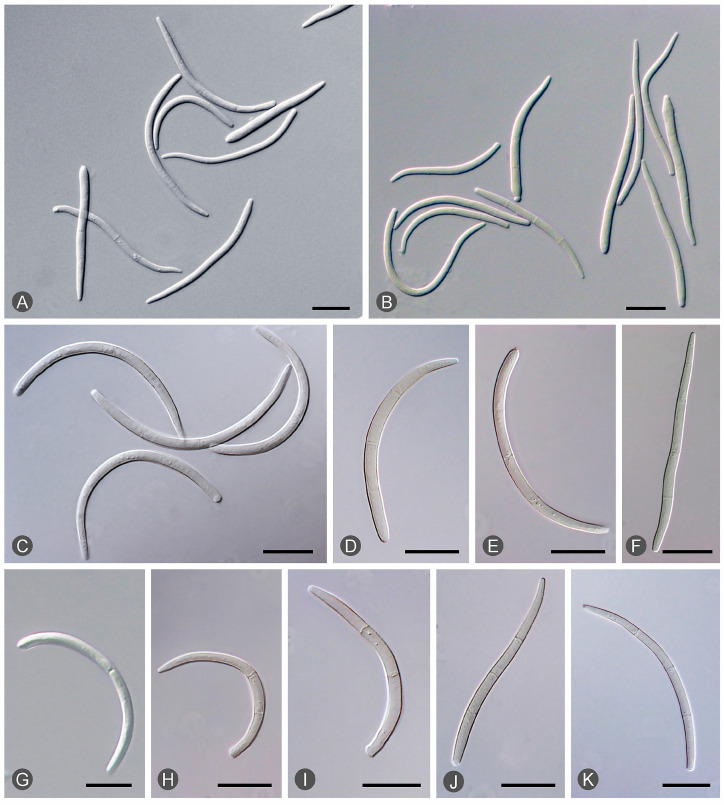
Morphological features of *Teratosphaeria destructans*: (**A**–**F**) curved and occasionally straight conidia with septa from different *Eucalyptus* trees. (**G**–**K**) Conidia with one to three septa: one septum (**G**); two septa (**H**,**I**); and three septa (**J**,**K**).

**Figure 5 microorganisms-12-00129-f005:**
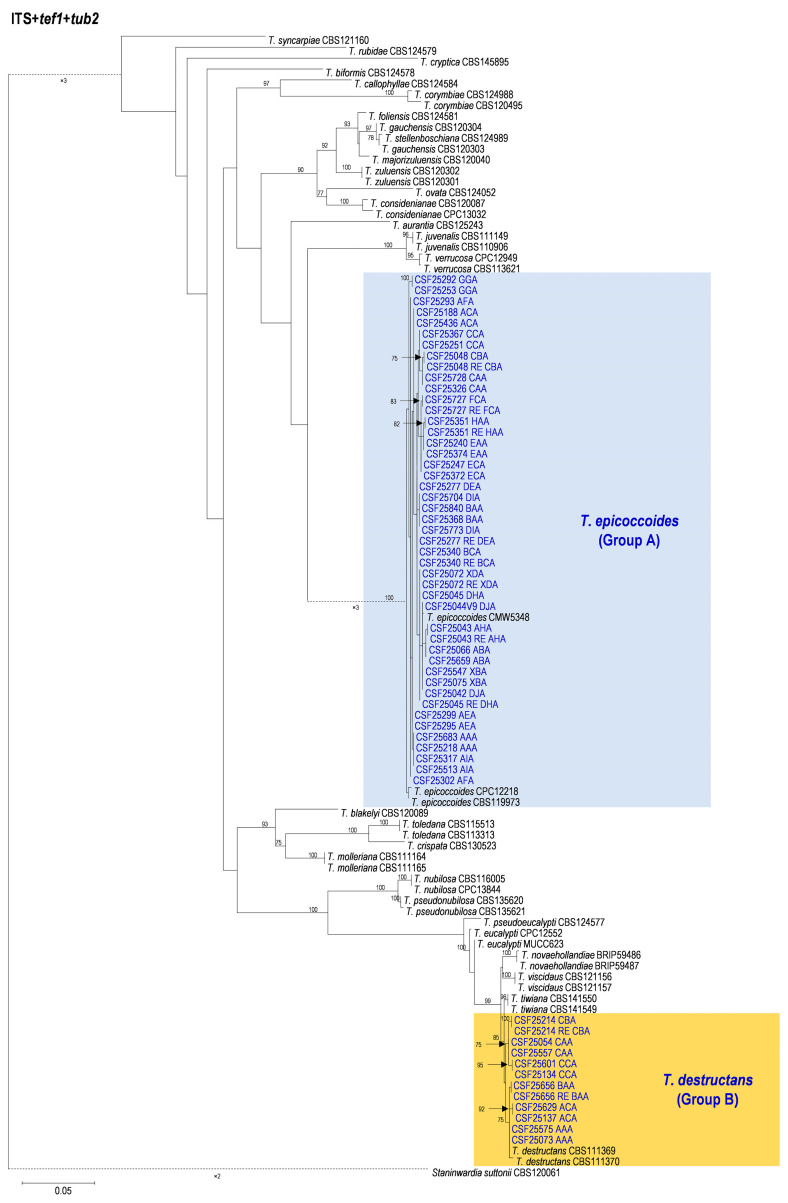
Phylogenetic tree obtained from the maximum likelihood (ML) analysis of the combined dataset of ITS, *tef1*, and *tub2* in these three gene regions. Bootstrap values ≥ 75% from the ML analysis are indicated at nodes. Isolates reported in this study are highlighted in blue.

**Figure 6 microorganisms-12-00129-f006:**
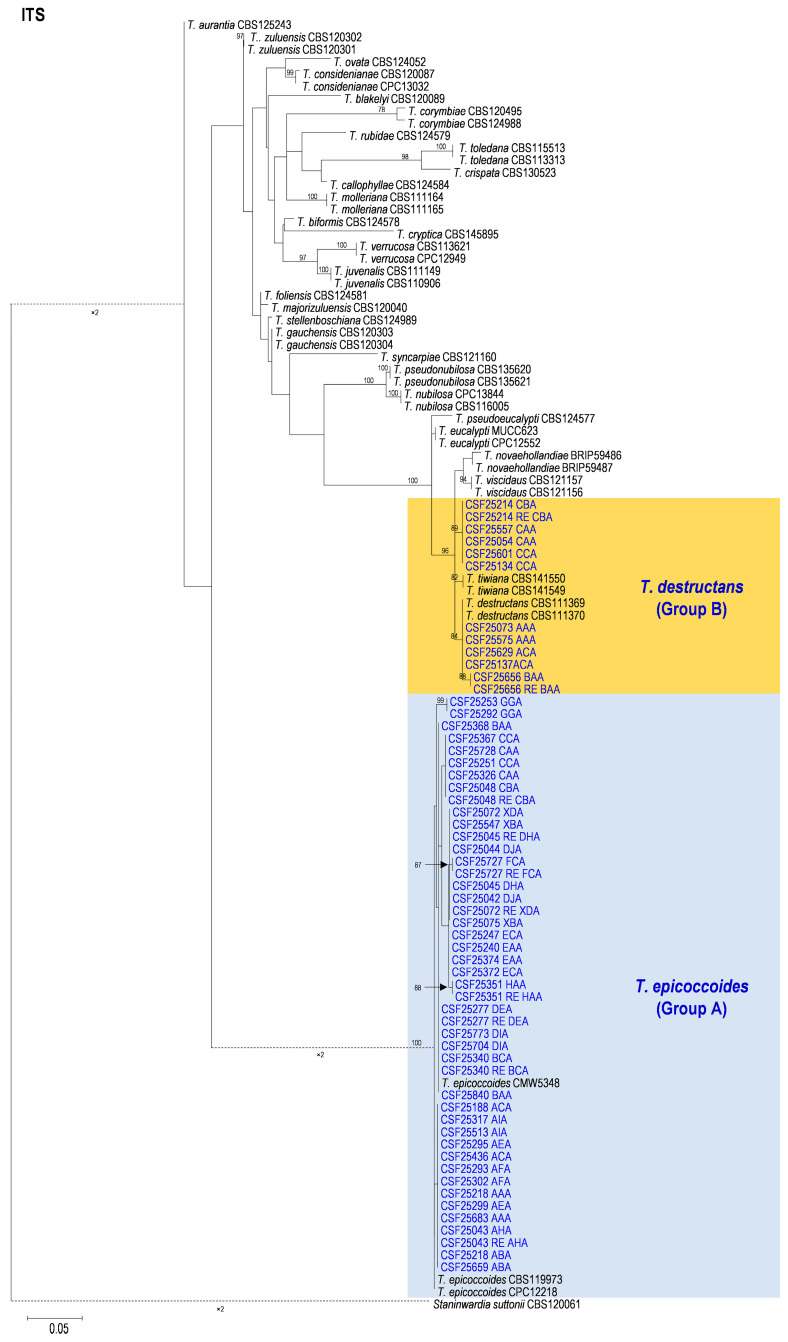
Phylogenetic tree obtained from the maximum likelihood (ML) analysis of the dataset of the ITS region. Bootstrap values ≥ 75% from the ML analysis are indicated at nodes. Isolates reported in this study are highlighted in blue.

**Figure 7 microorganisms-12-00129-f007:**
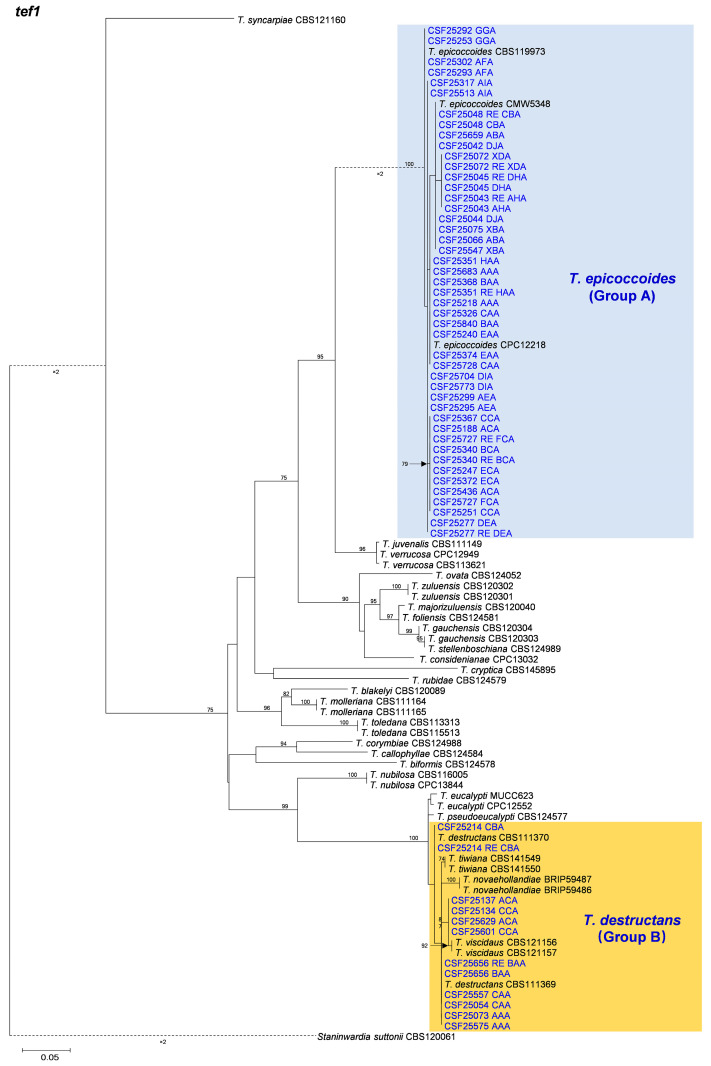
Phylogenetic tree obtained from the maximum likelihood (ML) analysis of the dataset of the *tef1* region. Bootstrap values ≥ 75% from the ML analysis are indicated at nodes. Isolates reported in this study are highlighted in blue.

**Figure 8 microorganisms-12-00129-f008:**
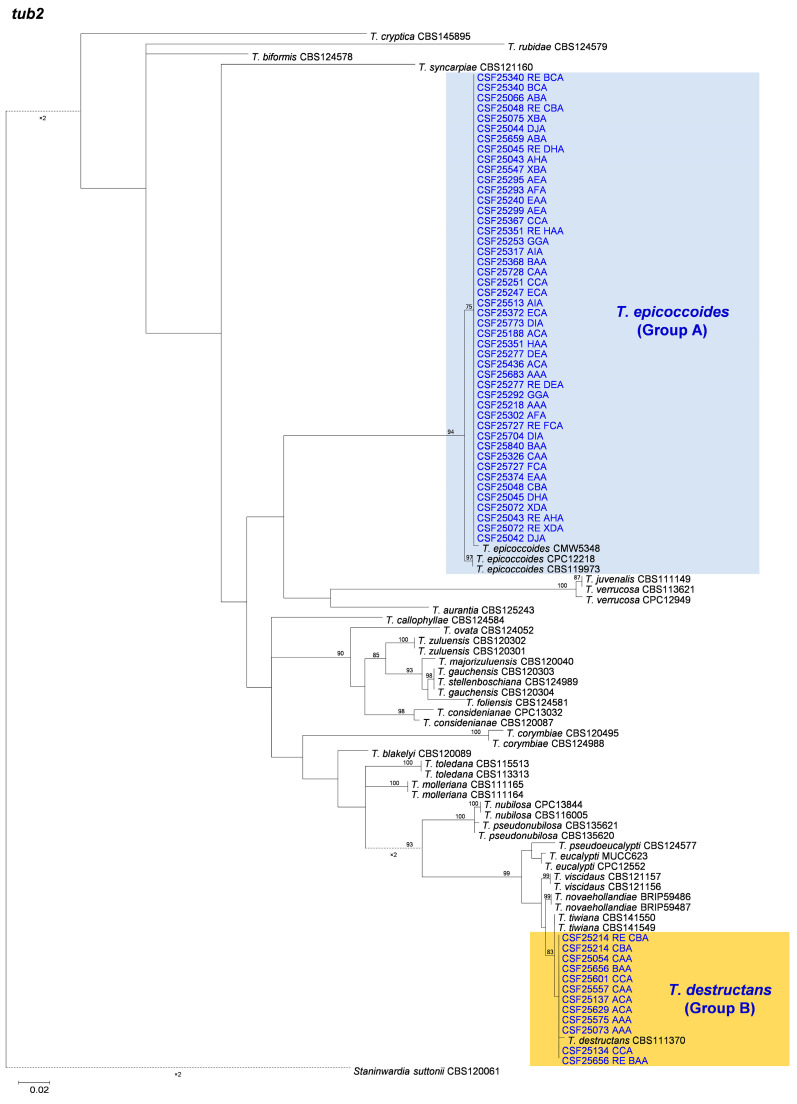
Phylogenetic tree obtained from the maximum likelihood (ML) analysis of the dataset of the *tub2* region. Bootstrap values ≥ 75% from the ML analysis are indicated at nodes. Isolates reported in this study are highlighted in blue.

**Table 1 microorganisms-12-00129-t001:** Location details, collection information, and *Eucalyptus* tree plantation species of diseased leaf samples collected from 59 sites in five provinces.

Sampling Site No. ^a^	Sample and Isolate Information ^b^	Province	Location Details	GPS Information	Hosts	Collector	Identified *Teratosphaeria* spp.	Sequenced Isolate Number	Identified Isolate Number of *T. epicoccoides*	Identified Isolate Number of *T. destructans*
1	20230315-2	Yunnan	Xin Village, Yongping Town, Jinggu County, Puer Region, Yunnan	23°19′44.6052″ N, 100°21′40.014″ E	2-year-old *E. urophylla* × *E. grandis*	B.Y. Chen and L.Q. Lu	*T. epicoccoides* and *T. destructans*	14	7	7
2	20230315-1	Yunnan	Hedong Village, Weiyuan Town, Jinggu County, Puer Region, Yunnan	23°30′52.0056″ N, 100°40′52.4568″ E	1-year-old *E. urophylla*	B.Y. Chen and L.Q. Lu	*T. epicoccoides* and *T. destructans*	16	2	14
3	20230314-2	Yunnan	Baomu Village, Fengshan Town, Jinggu County, Puer Region, Yunnan	23°43′47.6904″ N, 100°50′40.3008″ E	2-year-old *E. urophylla* × *E. tereticomis*	B.Y. Chen and L.Q. Lu	*T. epicoccoides*	6	6	0
4	20230314-1	Yunnan	Xin Village, Anban Town, Zhenyuan County, Puer Region, Yunnan	23°53′15.0918″ N, 100°55′25.1508″ E	3-year-old *E. urophylla* × *E. pellita*	B.Y. Chen and L.Q. Lu	*T. destructans*	3	0	3
5	20230315-3	Yunnan	Dawangtian Village, Ninger Town, Ninger County, Puer Region, Yunnan	23°9′51.0228″ N, 100°59′48.1596″ E	1-year-old *E. urophylla* × *E. grandis*	B.Y. Chen and L.Q. Lu	*T. epicoccoides* and *T. destructans*	14	5	9
6	20230316-2	Yunnan	Longjing Village, Gejiu County, Honghe Region, Yunnan	23°25′4.7496″ N, 103°14′21.9012″ E	2-year-old *E. urophylla* × *E. grandis*	B.Y. Chen and L.Q. Lu	*T. epicoccoides*	6	6	0
7	20230317-1	Yunnan	Chachang Village, Yangliujing Town, Guangnan County, Wenshan Region, Yunnan	23°56′26.412″ N, 105°15′38.9916″ E	2-year-old *E. urophylla* × *E. grandis*	B.Y. Chen and L.Q. Lu	*T. epicoccoides*	6	6	0
8	20230318-1	Yunnan	Administration center of Xinhua Town, Funing County, Wenshan Region, Yunnan	23°36′43.11″ N, 105°37′7.1256″ E	2-year-old *E. urophylla* × *E. grandis*	B.Y. Chen and L.Q. Lu	*T. epicoccoides*	6	6	0
9	20230318-2	Yunnan	Podi Village, Xinhua Town, Funing County, Wenshan Region, Yunnan	23°34′57.3564″ N, 105°34′34.7016″ E	1-year-old *E. urophylla* × *E. grandis*	B.Y. Chen and L.Q. Lu	*T. epicoccoides*	6	6	0
10	20230318-3	Yunnan	Nahong Village, Zhesang Town, Funing County, Wenshan Region, Yunnan	23°44′40.1316″ N, 105°57′12.204″ E	1-year-old *E. urophylla* × *E. grandis*	B.Y. Chen and L.Q. Lu	*T. epicoccoides*	6	6	0
11	20230319-1	Yunnan	Zhening Village, Boai Town, Funing County, Wenshan Region, Yunnan	23°49′53.4504″ N, 106°1′1.4196″ E	2-year-old *E. urophylla* × *E. tereticomis*	B.Y. Chen and L.Q. Lu	*T. epicoccoides*	5	5	0
12	20220703-1	Guangxi	Xichang Village, Liuqiao Town, Fusui County, Chongzuo Region, Guangxi	22°17′59.0496″ N, 107°43′39.936″ E	2-year-old *E. urophylla* × *E. grandis*	S.F. Chen, W.X. Wu, S.M. Xiang, B.Y. Chen and L.Q. Lu	*T. epicoccoides* and *T. destructans*	12	7	5
13	20220703-3	Guangxi	Jiucheng Village, Dongmen Town, Fusui County, Chongzuo Region, Guangxi	22°19′18.4152″ N, 107°47′24.9792″ E	0.5-year-old *E. urophylla* × *E. grandis*	S.F. Chen, W.X. Wu, S.M. Xiang, B.Y. Chen and L.Q. Lu	*T. epicoccoides* and *T. destructans*	15	7	8
14	20220703-2	Guangxi	Namengxin Village, Dongmen Town, Fusui County, Chongzuo Region, Guangxi	22°20′39.4476″ N, 107°52′20.9568″ E	2-year-old *E. urophylla* × *E. grandis*	S.F. Chen, W.X. Wu, S.M. Xiang, B.Y. Chen and L.Q. Lu	*T. epicoccoides* and *T. destructans*	13	6	7
15	20220703-4	Guangxi	Chongwen Village, Dazhi Town, Qinbei District, Qinzhou Region, Guangxi	22°1′52.8816″ N, 108°21′27.9576″ E	0.5-year-old *E. urophylla* × *E. grandis*	S.F. Chen, W.X. Wu, S.M. Xiang, B.Y. Chen and L.Q. Lu	*T. epicoccoides* and *T. destructans*	10	2	8
16	20220702-3	Guangxi	Youxing Village, Dongchang Town, Qinnan District, Qinzhou Region, Guangxi	21°53′7.4832″ N, 108°46′15.3516″ E	0.5-year-old *E. urophylla* × *E. grandis*	S.F. Chen, W.X. Wu, S.M. Xiang, B.Y. Chen and L.Q. Lu	*T. epicoccoides* and *T. destructans*	14	3	11
17	20220702-2	Guangxi	Bengtang Village, Nali Town, Qinnan District, Qinzhou Region, Guangxi	21°51′37.368″ N, 108°55′57.648″ E	1-year-old *E. urophylla* × *E. grandis*	S.F. Chen, W.X. Wu, S.M. Xiang, B.Y. Chen and L.Q. Lu	*T. epicoccoides* and *T. destructans*	14	5	9
18	20220702-1	Guangxi	Dama Village, Nali Town, Qinnan District, Qinzhou Region, Guangxi	21°50′39.5124″ N, 108°55′55.3404″ E	0.5-year-old *E. urophylla* × *E. pellita*	S.F. Chen, W.X. Wu, S.M. Xiang, B.Y. Chen and L.Q. Lu	*T. epicoccoides* and *T. destructans*	14	8	6
19	20220704-1	Guangxi	Chezouling Village, Fucheng Town, Yinhai District, Beihai Region, Guangxi	21°38′8.7252″ N, 109°17′51.4644″ E	0.5-year-old *E. urophylla* × *E. grandis*	S.F. Chen, W.X. Wu, S.M. Xiang, B.Y. Chen and L.Q. Lu	*T. epicoccoides* and *T. destructans*	15	5	10
20	20220704-2	Guangxi	Wuliu Village, Shankou Town, Hepu County, Beihai Region, Guangxi	21°33′10.3248″ N, 109°42′43.2324″ E	1-year-old *E. urophylla* × *E. pellita*	S.F. Chen, W.X. Wu, S.M. Xiang, B.Y. Chen and L.Q. Lu	*T. epicoccoides* and *T. destructans*	20	6	14
21	20220704-3	Guangxi	Gaopo Village, Shankou Town, Hepu County, Beihai Region, Guangxi	21°34′3.9792″ N, 109°43′8.6304″ E	2-year-old *E. urophylla*	S.F. Chen, W.X. Wu, S.M. Xiang, B.Y. Chen and L.Q. Lu	*T. epicoccoides* and *T. destructans*	16	6	10
22	20221026-6	Hainan	Administration center of Dongying Town, Lingao County, Hainan	19°57′58.788″ N, 109°38′43.206″ E	2-year-old *E. urophylla* × *E. grandis*	S.F. Chen, Y. Liu, X.Y. Liang, S.M. Xiang, B.Y. Chen and L.Q. Lu	*T. epicoccoides*	3	3	0
23	20221026-7	Hainan	Wenshan Village, Dongying Town, Lingao County, Hainan	19°58′36.4656″ N, 109°38′53.6352″ E	1-year-old *E. urophylla* × *E. grandis*	S.F. Chen, Y. Liu, X.Y. Liang, S.M. Xiang, B.Y. Chen and L.Q. Lu	*T. epicoccoides* and *T. destructans*	17	6	11
24	20221026-4	Hainan	Meisheng Village, Lincheng Town, Lingao County, Hainan	19°46′16.644″ N, 109°41′31.74″ E	2-year-old *E. urophylla* × *E. grandis*	S.F. Chen, Y. Liu, X.Y. Liang, S.M. Xiang, B.Y. Chen and L.Q. Lu	*T. epicoccoides* and *T. destructans*	10	6	4
25	20221026-5	Hainan	Lanluo Village, Lincheng Town, Lingao County, Hainan	19°47′49.0452″ N, 109°41′8.7324″ E	2 to 3-year-old *E. urophylla* × *E. tereticomis*	S.F. Chen, Y. Liu, X.Y. Liang, S.M. Xiang, B.Y. Chen and L.Q. Lu	*T. epicoccoides* and *T. destructans*	15	5	10
26	20221026-2	Hainan	Dunxiang Village, Jialai Town, Lingao County, Hainan	19°43′34.8276″ N, 109°42′52.1712″ E	2-year-old *E. urophylla* × *E. grandis*	S.F. Chen, Y. Liu, X.Y. Liang, S.M. Xiang, B.Y. Chen and L.Q. Lu	*T. epicoccoides* and *T. destructans*	9	6	3
27	20221026-3	Hainan	Duolang Village, Duowen Town, Lingao County, Hainan	19°45′45.1548″ N, 109°44′28.518″ E	2-year-old *E. urophylla* × *E. tereticomis*	S.F. Chen, Y. Liu, X.Y. Liang, S.M. Xiang, B.Y. Chen and L.Q. Lu	*T. destructans*	7	0	7
28	20221026-1	Hainan	Songbai Village, Huangtong Town, Lingao County, Hainan	19°47′41.46″ N, 109°49′52.7196″ E	1-year-old *E. urophylla* × *E. grandis*	S.F. Chen, Y. Liu, X.Y. Liang, S.M. Xiang, B.Y. Chen and L.Q. Lu	*T. epicoccoides*	5	5	0
29	20230221-2	Guangdong	Fanchang Village, Longmen Town, Leizhou County, Zhanjiang Region, Guangdong	20°40′3.018″ N, 110°1′50.358″ E	1-year-old *E. urophylla* × *E. grandis*	W.X. Wu, X.Y. Liang, B.Y. Chen and L.Q. Lu	*T. epicoccoides* and *T. destructans*	18	6	12
30	20230221-1	Guangdong	Naqi Village, Chengbei Town, Xuwen County, Zhanjiang Region, Guangdong	20°20′49.56″ N, 110°9′58.5864″ E	1-year-old *E. urophylla* × *E. grandis*	W.X. Wu, X.Y. Liang, B.Y. Chen and L.Q. Lu	*T. epicoccoides* and *T. destructans*	19	6	13
31	20230220-3	Guangdong	Houkeng Village, Chengyue Town, Leizhou County, Zhanjiang Region, Guangdong	21°7′53.886″ N, 110°6′4.932″ E	2-year-old *E. urophylla* × *E. grandis*	W.X. Wu, X.Y. Liang, B.Y. Chen and L.Q. Lu	*T. epicoccoides* and *T. destructans*	18	6	12
32	20230221-3	Guangdong	Lizhikeng Village, Shicheng Town, Lianjiang County, Zhanjiang Region, Guangdong	21°31′37.7292″ N, 110°16′12.648″ E	1-year-old *E. urophylla* × *E. grandis*	W.X. Wu, X.Y. Liang, B.Y. Chen and L.Q. Lu	*T. epicoccoides* and *T. destructans*	19	6	13
33	20230220-2	Guangdong	Xiachen Village, Rudong Town, Yangxi County, Yangjiang Region, Guangdong	21°36′43.29″ N, 111°26′23.118″ E	1-year-old *E. urophylla* × *E. grandis*	W.X. Wu, X.Y. Liang, B.Y. Chen and L.Q. Lu	*T. epicoccoides* and *T. destructans*	14	3	11
34	20230220-1	Guangdong	Li Village, Chengcun Town, Yangxi County, Yangjiang Region, Guangdong	21°49′47.5284″ N, 111°43′3.8964″ E	1-year-old *E. urophylla* × *E. grandis*	W.X. Wu, X.Y. Liang, B.Y. Chen and L.Q. Lu	*T. epicoccoides* and *T. destructans*	10	2	8
35	20230216-5	Guangdong	Xiadaidong Village, Huilong Town, Deqing County, Zhaoqing Region, Guangdong	23°11′35.2032″ N, 111°41′21.1596″ E	1-year-old *E. urophylla* × *E. grandis*	W.X. Wu, X.Y. Liang, B.Y. Chen and L.Q. Lu	*T. epicoccoides*	6	6	0
36	20230216-3	Guangdong	Daxintang Village, Helang Town, Yangchun County, Yangjiang Region, Guangdong	22°34′14.4336″ N, 111°51′53.928″ E	1-year-old *E. urophylla* × *E. grandis*	W.X. Wu, X.Y. Liang, B.Y. Chen and L.Q. Lu	*T. epicoccoides*	6	6	0
37	20230216-4	Guangdong	Xin Village, Helang Town, Yangchun County, Yangjiang Region, Guangdong	22°33′32.3748″ N, 111°51′22.4568″ E	1-year-old *E. urophylla* × *E. grandis*	W.X. Wu, X.Y. Liang, B.Y. Chen and L.Q. Lu	*T. epicoccoides*	4	4	0
38	20230216-2	Guangdong	Tangxia Village, Songbai Town, Yangchun County, Yangjiang Region, Guangdong	22°27′23.958″ N, 111°53′30.9192″ E	1-year-old *E. urophylla* × *E. grandis*	W.X. Wu, X.Y. Liang, B.Y. Chen and L.Q. Lu	*T. epicoccoides*	6	6	0
39	20230215-1	Guangdong	Xin Village, Shiwang Town, Yangchun County, Yangjiang Region, Guangdong	22°33′2.6928″ N, 111°57′34.7616″ E	1-year-old *E. urophylla* × *E. grandis*	W.X. Wu, X.Y. Liang, B.Y. Chen and L.Q. Lu	*T. epicoccoides*	6	6	0
40	20230216-1	Guangdong	Shuizhai Village, Chunwan Town, Yangchun County, Yangjiang Region, Guangdong	22°25′8.1876″ N, 111°56′20.4252″ E	1-year-old *E. urophylla* × *E. grandis*	W.X. Wu, X.Y. Liang, B.Y. Chen and L.Q. Lu	*T. epicoccoides*	6	6	0
41	20230217-3	Guangdong	Fengji Village, Lixi Town, Yingde County, Qingyuan Region, Guangdong	23°55′21.4824″ N, 113°10′20.4816″ E	1-year-old *E. urophylla* × *E. grandis*	W.X. Wu, X.Y. Liang, B.Y. Chen and L.Q. Lu	*T. epicoccoides* and *T. destructans*	18	7	11
42	20230219-1	Guangdong	Kengwei Village, Feilaixia Town, Qingcheng District, Qingyuan Region, Guangdong	23°53′2.004″ N, 113°10′4.2456″ E	1-year-old *E. urophylla* × *E. grandis*	W.X. Wu, X.Y. Liang, B.Y. Chen and L.Q. Lu	*T. destructans*	10	0	10
43	20230217-2	Guangdong	Xikeng Village, Feilaixia Town, Qingcheng District, Qingyuan Region, Guangdong	23°50′40.7112″ N, 113°9′38.2932″ E	1-year-old *E. urophylla* × *E. grandis*	W.X. Wu, X.Y. Liang, B.Y. Chen and L.Q. Lu	*T. epicoccoides*	6	6	0
44	20230217-1	Guangdong	Donger Village, Dongcheng Subdistrict, Qingcheng District, Qingyuan Region, Guangdong	23°45′27.1476″ N, 113°8′16.656″ E	1-year-old *E. urophylla* × *E. grandis*	W.X. Wu, X.Y. Liang, B.Y. Chen and L.Q. Lu	*T. epicoccoides*	6	6	0
45	20230218-1	Guangdong	Huangwuxin Village, Maba Town, Qujiang District, Shaoguan Region, Guangdong	24°37′49.3284″ N, 113°35′10.8384″ E	2-year-old *E. urophylla* × *E. grandis*	W.X. Wu, X.Y. Liang, B.Y. Chen and L.Q. Lu	*T. epicoccoides*	6	6	0
46	20230217-4	Guangdong	Heping Village, Wangbu Town, Yingde County, Qingyuan Region, Guangdong	24°11′49.0884″ N, 113°28′19.1424″ E	2-year-old *E. urophylla* × *E. grandis*	W.X. Wu, X.Y. Liang, B.Y. Chen and L.Q. Lu	*T. epicoccoides*	6	6	0
47	20230218-2	Guangdong	Xinzouwu Village, Wushi Town, Qujiang District, Shaoguan Region, Guangdong	24°36′12.618″ N, 113°38′22.0128″ E	1-year-old *E. urophylla* × *E. grandis*	W.X. Wu, X.Y. Liang, B.Y. Chen and L.Q. Lu	*T. epicoccoides*	6	6	0
48	20230218-3	Guangdong	Yiwu Village, Datang Town, Qujiang District, Shaoguan Region, Guangdong	24°43′37.956″ N, 113°42′49.9536″ E	1-year-old *E. urophylla* × *E. grandis*	W.X. Wu, X.Y. Liang, B.Y. Chen and L.Q. Lu	*T. epicoccoides*	6	6	0
49	20230218-4	Guangdong	Xin Village, Daqiao Town, Renhua County, Shaoguan Region, Guangdong	24°55′28.0596″ N, 113°46′30.2232″ E	2-year-old *E. urophylla* × *E. grandis*	W.X. Wu, X.Y. Liang, B.Y. Chen and L.Q. Lu	*T. epicoccoides*	6	6	0
50	20230330-1	Fujian	Qing Village, Ansha Town, Yongan County, Sanming Region, Fujian	25°59′56.4864″ N, 117°6′47.772″ E	2-year-old *E. urophylla* × *E. pellita*	B.Y. Chen and L.Q. Lu	*T. epicoccoides*	5	5	0
51	20230328-1	Fujian	Zhanglin Village, Caoyuan Town, Yongan County, Sanming Region, Fujian	26°1′34.122″ N, 117°18′6.5556″ E	2-year-old *E. urophylla* × *E. grandis*	B.Y. Chen and L.Q. Lu	*T. epicoccoides*	6	6	0
52	20230329-1	Fujian	Linggan Village, Dahu Town, Yongan County, Sanming Region, Fujian	26°4′45.6024″ N, 117°19′15.186″ E	7-year-old *E. urophylla* × *E. grandis*	B.Y. Chen and L.Q. Lu	*T. epicoccoides*	6	6	0
53	20230329-2	Fujian	Wenbeixin Village, Caoyuan Town, Yongan County, Sanming Region, Fujian	26°0′45.684″ N, 117°18′44.964″ E	1-year-old *E. urophylla* × *E. pellita*	B.Y. Chen and L.Q. Lu	*T. epicoccoides*	6	6	0
54	20230331-3	Fujian	Hekeng Village, Guantian Town, Zhangping County, Longyan Region, Fujian	25°1′7.4604″ N, 117°29′14.9388″ E	2-year-old *E. urophylla* × *E. grandis*	B.Y. Chen and L.Q. Lu	*T. epicoccoides*	6	6	0
55	20230331-2	Fujian	Xipo Village, Hulin Town, Huaan County, Zhangzhou Region, Fujian	25°7′1.6356″ N, 117°32′59.4276″ E	1-year-old *E. urophylla* × *E. grandis*	B.Y. Chen and L.Q. Lu	*T. epicoccoides*	6	6	0
56	20230331-1	Fujian	Shangxue Village, Huafeng Town, Huaan County, Zhangzhou Region, Fujian	24°57′58.5504″ N, 117°34′0.5592″ E	3-year-old *E. urophylla* × *E. pellita*	B.Y. Chen and L.Q. Lu	*T. epicoccoides*	6	6	0
57	20230330-3	Fujian	Dakeng Village, Shajian Town, Huaan County, Zhangzhou Region, Fujian	24°45′37.134″ N, 117°35′51.2052″ E	1-year-old *E. urophylla* × *E. grandis*	B.Y. Chen and L.Q. Lu	*T. epicoccoides*	6	6	0
58	20230330-2	Fujian	Xiawenkeng Village, Yanxi Town, Changtai District, Zhangzhou Region, Fujian	24°45′43.3476″ N, 117°48′4.986″ E	2-year-old *E. urophylla* × E. tereticomis	B.Y. Chen and L.Q. Lu	*T. epicoccoides*	6	6	0
59	20230401-1	Fujian	Dudong Village, Changtai District, Zhangzhou Region, Fujian	24°37′52.3956″ N, 117°42′12.3624″ E	2-year-old *E. urophylla* × *E. pellita*	B.Y. Chen and L.Q. Lu	*T. epicoccoides*	6	6	0

^a^ Code of 59 sampling sites connecting to “Sampling Site No.” in Table 2. ^b^ Information associated with sample site and isolate, for example, “20230315-2” indicates sampling site “20230315-2” and isolate from this site; the sampling site number connecting to “Sample and Isolate Information” in Table 2.

**Table 3 microorganisms-12-00129-t003:** Isolates from other studies used in phylogenetic analyses in this study.

Species	Isolate No. ^a^	Type	Hosts	Location	Collector	GenBank Accession No. ^b^			References or Source of Data
						ITS	*tef1*	*tub2*	
*Teratosphaeria aurantia*	CBS 125243 = MUCC 668	ex-type	*Eucalyptus grandis*	Queensland, Australia	G. Whyte	KF901561	N/A ^c^	KF902984	[30]
*T. biformis*	CBS 124578 = MUCC 693	ex-type	*Eucalyptus globulus*	Queensland, Australia	G. Whyte	KF901564	KF903287	KF902987	[30]
*T. blakelyi*	CBS 120089 = CPC 12837	ex-type	*Eucalyptus blakelyi*	New South Wales, Australia	B.A. Summerell	KF901565	KF903288	KF902988	[30]
*T. callophyllae*	CBS 124584 = MUCC 700	ex-type	*Corymbia calophylla*	Western Australia, Australia	K. Taylor	KF901566	KF903289	FJ532003	[23,30]
*T. considenianae*	CBS 120087 = CPC 12940	ex-type	*Eucalyptus consideniana*	New South Wales, Australia	B.A. Summerell	DQ923527	N/A	FJ952510	[23,30]
	CPC 13032	unknown	*Eucalyptus* sp.	New South Wales, Australia	B.A. Summerell	KF901567	KF903291	KF902990	[30]
*T. corymbiae*	CBS 120495 = DAR 77445	ex-type	*Corymbia maculata*	New South Wales, Australia	A.J. Carnegie	EF011657	N/A	FJ532005	[23]
	CBS 124988 = CPC 13125	unknown	*Corymbia henryi*	New South Wales, Australia	A.J. Carnegie	KF901569	KF903293	KF902992	[30]
*T. crispata*	CBS 130523	ex-type	*Eucalyptus bridgesiana*	New South Wales, Australia	A.J. Carnegie	GQ890348	N/A	N/A	[23,30]
*T. cryptica*	CBS 145895 = CPC 13839	ex-epitype	*Eucalyptus globulus*	Australia	I.W. Smith	KF901572	KF903298	KF902997	[23,30]
*T. destructans*	CBS 111369 = CMW 5219 = CPC 1366	ex-type	*Eucalyptus grandis*	Indonesia	M.J. Wingfield	DQ267595	DQ235113	N/A	[23]
	CBS 111370 = CPC 1368	ex-type	*Eucalyptus grandis*	Indonesia	M.J. Wingfield	KF901574	KF903301	KF903000	[23,30]
*T. epicoccoides*	CMW 5348 = CPC 1346	ex-type	*Eucalyptus* sp.	Indonesia	M.J. Wingfield	DQ239972	DQ240170	DQ240117	[23]
	CBS 119973	unknown	*Eucalyptus pellita*	Vietnam	T.I. Burgess	KF901784	KF903359	KF903055	[30]
	CPC 12218	unknown	*Eucalyptus* sp.	Indonesia	M.J. Wingfield	KF901664	KF903357	KF903054	[30]
*T. eucalypti*	CPC 12552	not type	*Eucalyptus nitens*	Tasmania, Australia	C. Mohammed	KF901576	KF903303	KF903002	[30]
	MUCC 623	not type	*Eucalyptus nitens*	New South Wales, Australia	A.J. Carnegie	FJ793255	EU101623	EU101566	[25]
*T. foliensis*	CBS 124581 = MUCC 670	ex-type	*Eucalyptus globulus*	New South Wales, Australia	S. Collins	KF901580	KF903311	KF903009	[30]
*T. gauchensis*	CBS 120303 = CMW 17331	ex-type	*Eucalyptus grandis*	Uruguay	M.J. Wingfield	KF901790	KF903315	KF903013	[30]
	CBS 120304 = CMW 17332	ex-type	*Eucalyptus grandis*	Uruguay	M.J. Wingfield	KF901789	KF903314	KF903012	[30]
*T. juvenalis*	CBS 110906 = CMW 13347	ex-type	*Eucalyptus cladocalyx*	South Africa	P.W. Crous	KF901730	N/A	N/A	[30]
	CBS 111149	unknown	*Eucalyptus cladocalyx*	South Africa	P.W. Crous	KF901670	KF903317	KF903015	[30]
*T. majorizuluensis*	CBS 120040 = CPC 12712	ex-type	*Eucalyptus botryoides*	New South Wales, Australia	B.A. Summerell	KF901581	KF903319	KF903017	[30]
*T. molleriana*	CBS 111164 = CMW 4940 = CPC 1214	ex-type	*Eucalyptus globulus*	Portugal	S. McCrae	KF901692	KF903324	KF903021	[30]
	CBS 111165 = CPC 1215	ex-type	*Eucalyptus globulus*	Portugal	S. McCrae	KF901693	KF903325	KF903022	[30]
*T. novaehollandiae*	BRIP 59486	ex-type	*Eucalyptus camaldulensis*	Western Australia, Australia	A. Maxwell	KT972281	KT972345	KT972313	[25]
	BRIP 59487	not type	*Eucalyptus camaldulensis*	Western Australia, Australia	A. Maxwell	KT972282	KT972346	KT972314	[25]
*T. nubilosa*	CBS 116005 = CMW 3282 = CPC 937	ex-epitype	*Eucalyptus globulus*	Victoria, Australia	A.J. Carnegie	KF901686	KF903336	KF903033	[23,30]
	CPC 13844	not type	*Eucalyptus globulus*	Australia	I.W. Smith	KF901590	KF903344	KF903041	[30]
*T. ovata*	CBS 124052 = CPC 14632	unknown	*Eucalyptus phoenicea*	New South Wales, Australia	B.A. Summerell	KF901591	KF903345	KF903042	[30]
*T. pseudoeucalypti*	CBS 124577 = MUCC 607	ex-type	*Eucalyptus grandis* × *E. camaldulensis*	Queensland, Australia	G.S. Pegg	KF901593	KF903349	KF903046	[30]
*T. pseudonubilosa*	CBS 135621 = CMW 30745	ex-type	*Eucalyptus globulus*	Victoria, Australia	G. Pérez	HQ130818	N/A	HQ131318	[23]
	CBS 135620 = CMW 30723	link to paratype	*Eucalyptus globulus*	Western Australia, Australia	G. Pérez	HQ130817	N/A	HQ131317	[23]
*T. rubidae*	CBS 124579 = MUCC 658	ex-type	*Corymbia calophylla*	Western Australia, Australia	P.A. Barber	KF901596	KF903352	KF903049	[30]
*T. stellenboschiana*	CBS 124989 = CPC 13767	unknown	*Eucalyptus punctata*	South Africa	P.W. Crous and G. Bills	KF901732	KF903355	KF903052	[30]
*T. syncarpiae*	CBS 121160 = DAR 77433	ex-type	*Syncarpia glomulifera*	New South Wales, Australia	A.J. Carnegie and M.J. Wingfield	KF901598	KF903360	KF903056	[30]
*T. tiwiana*	CBS 141549 = BRIP 63496	ex-type	*Eucalyptus urophylla hybrids*	Tiwi Island, Australia	T.I. Burgess	KT972297	KT972362	KT972330	[25]
	CBS 141550 = BRIP 63497	not type	*Eucalyptus urophylla hybrids*	Tiwi Island, Australia	T.I. Burgess	KT972305	KT972369	KT972337	[25]
*T. toledana*	CBS 113313 = CMW 14457	ex-type	*Eucalyptus* sp.	Spain	P.W. Crous	KF901734	KF903361	KF903058	[30]
	CBS 115513 = CPC 10840	unknown	*Eucalyptus* sp.	Spain	P.W. Crous and G. Bills	KF901600	KF903362	KF903059	[30]
*T. verrucosa*	CBS 113621 = CPC 42	ex-type	*Eucalyptus cladocalyx*	South Africa	P.W. Crous	KF901645	KF903365	KF903062	[30]
	CPC 12949	unknown	*Eucalyptus* sp.	South Africa	P.W. Crous	KF901768	KF903364	KF903061	[30]
*T. viscida*	CBS 121157 = MUCC 453	ex-type	*Eucalyptus grandis*	Mareeba, Australia	T.I. Burgess	EF031472	EF031496	EF031484	[25]
	CBS 121156 = MUCC 452	not type	*Eucalyptus grandis*	Mareeba, Australia	T.I. Burgess	EF031471	EF031495	EF031483	[25]
*T. zuluensis*	CBS 120301 = CMW 17321	ex-epitype	*Eucalyptus grandis*	South Africa	M.J. Wingfield	KF901735	KF903368	KF903064	[30]
	CBS 120302 = CMW 17322	ex-epitype	*Eucalyptus grandis*	South Africa	M.J. Wingfield	KF901736	KF903369	KF903065	[30]
*Staninwardia suttonii*	CBS 120061 = CPC 13055	ex-type	*Eucalyptus robusta*	Australia	B.A. Summerell	KF901552	KF903270	KF902974	[30]

^a^ BRIP: fungal collection of Queensland Plant Pathology Herbarium (BRIP), Brisbane, Queensland, Australia; CBS: Westerdijk Fungal Biodiversity Institute, Utrecht, The Netherlands; CMW: Culture collection of the Forestry and Agricultural Biotechnology Institute (FABI), University of Pretoria, Pretoria, South Africa; CPC: Pedro Crous working collection housed at Westerdijk Fungal Biodiversity Institute; DAR: Plant Pathology Herbarium, Orange Agricultural Institute, Forest Road, Orange. NSW 2800, Australia; MUCC: Murdoch University Culture Collection, Murdoch, Australia. ^b^ ITS = internal transcribed spacer regions and intervening 5.8S nrRNA gene; *tef1* = translation elongation factor 1-alpha gene; *tub2* = β-tubulin gene. ^c^ N/A: information not available.

**Table 4 microorganisms-12-00129-t004:** Isolate numbers of each genotype of *T. epicoccoides* in the *Eucalyptus* plantations in each of the five provinces.

Genotype ^a^	Yunnan	Guangxi	Guangdong	Hainan	Fujian	Five Provinces
AAA	19	5	21	3	9	57
ABA	1	2	0	0	0	3
ACA	5	13	18	4	4	44
AEA	0	1	0	3	0	4
AFA	0	1	0	2	0	3
AHA	0	1	0	0	0	1
AIA	0	0	1	2	0	3
BAA	0	1	4	1	2	8
BCA	0	0	1	0	0	1
CAA	3	2	7	2	5	19
CBA	0	1	0	0	0	1
CCA	0	1	2	0	3	6
DEA	0	0	0	1	0	1
DHA	0	1	0	0	0	1
DIA	6	1	0	2	9	18
DJA	0	2	0	0	0	2
EAA	16	11	35	3	17	82
ECA	4	6	14	2	7	33
FCA	1	0	0	0	0	1
GGA	0	1	0	1	0	2
HAA	0	0	1	0	0	1
Number of Genotype	8	16	10	12	8	21
Number of Isolate	55	50	104	26	56	291
Ratio of Number of Genotype to Number of Isolate	0.15	0.32	0.096	0.46	0.14	0.072

^a^ The genotype was determined by ITS-*tef1*-*tub2* gene sequences.

**Table 5 microorganisms-12-00129-t005:** Isolate numbers of each genotype of *T. destructans* in the *Eucalyptus* plantations in each of the five provinces.

Genotype ^a^	Yunnan	Guangxi	Guangdong	Hainan	Fujian	Five Provinces
AAA	32	37	49	14	0	132
ACA	0	1	5	4	0	10
BAA	1	0	0	0	0	1
CAA	0	45	27	15	0	87
CBA	0	1	0	0	0	1
CCA	0	1	9	1	0	11
Number of Genotype	2	5	4	4	0	6
Number of Isolate	33	85	90	34	0	242
Ratio of Number of Genotype to Number of Isolate	0.061	0.059	0.044	0.12	N/A ^b^	0.025

^a^ The genotype was determined by ITS-*tef1*-*tub2* gene sequences. ^b^ N/A indicates not available.

## Data Availability

Data are contained within the article.

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
