# Peer review of "Wide Distribution of Teratosphaeria epicoccoides and T. destructans Associated with Diseased Eucalyptus Leaves in Plantations in Southern China"

_microorganisms, 2024, doi:10.3390/microorganisms12010129_

Round 1
Reviewer 1 Report
Comments and Suggestions for Authors
The present manuscript describes a study on Teratosphaeria species associated with Eucalyptus trees in southern China. I think it was well designed and presents interesting results on leaf spots. After analysis, I raise the following points for the authors to use to improve the manuscript:
1. Figure 1 illustrates various aspects of Teratosphaeria leaf spots on Eucalyptus. I suggest that the resolution and quality of photos be improved to allow adequate visualization of symptoms and signs. I also recommend replacing the photo in Figure 1E, so there are no people. It would be interesting to add photos of the two types of conidia described for the two species of Teratosphaeria.
2. According to the text, what would be the likely origin of these species in the region studied? Could it have been endemic to the region or were the fungi introduced through storms or human action? If it was derived from human action, could it be linked to the issue of the introduction of infected seedlings into the region?
3. Has there been any assessment of severity in the different genetic materials of Eucalyptus, which could support the question of resistance assessment?
Author Response
Dear Reviewer,
Thanks for your comments and suggestions to improve our manuscript.
Please see our point by point responses from the attached document “Dec. 31, 2023_Response to Reviewer One Comments and Changes Made.docx”.
Thanks again and best wishes
ShuaiFei Chen and co-authors.

Reviewer 2 Report
Comments and Suggestions for Authors
The submitted manuscript deals with the current issue of the occurrence of pathogenic organisms and their spread outside the home area. Their determination, whether by classical or genetic methods, is current and necessary. For this reason, I believe that the article will be suitable for publication in this journal. I have the following notes and comments on the submitted manuscript. Here, the authors describe the occurrence of monitored pathogens on Eucalyptus trees. However, it is necessary to realize that this genus includes more than 800 species, therefore, based on the botanical nomenclature, the observed species should also be added. I believe there are certainly differences between species, varieties and hybrids. In the introductory part, I somewhat lack information about the occurrence of the relevant pathogens in Australia and the adjacent geographical area, i.e. in the homeland of Eucalyptus. The methodology does not include the course of the weather, as this characteristic is necessary with regard to the spread of pathogens. The characteristics of the plant material used are also missing. The tables in the results section are confusing and extensive. I recommend their modification. Similarly, graphs 3 to 6 are somewhat less clear. The discussion is sometimes descriptive. The authors draw on 62 literary sources, of which only 1x is from 2022 and 1x from 2019. The other sources are older. It is necessary to focus on more recent data.
Author Response
Dear Reviewer,
Thanks for your comments and suggestions to improve our manuscript.
Please see our point by point responses from the attached document “Dec. 31, 2023_Response to Reviewer Two Comments and Changes Made.docx”.
Thanks again and best wishes
ShuaiFei Chen and co-authors.

Round 2
Reviewer 2 Report
Comments and Suggestions for Authors The authors have submitted a revised version of the original manuscript, where this version is based on the reviewers' comments. Based on the evaluation of the original manuscript and the comparison of both versions, I can state that the authors accepted the comments and added the comments to the text. Furthermore, questions were answered and the answers to them were part of not only the authors' comments, but also a separate manuscript.